# The Signal is in the Steps: Local Scoring for Reasoning Data Selection

**Hoang Anh Just** [1]  **Myeongseob Ko** [1]  **Ruoxi Jia** [1]

## Abstract

Distilling long-form reasoning from teacher models into smaller students requires selecting which candidate solutions to train on. Recent work argues that one should select responses the student model assigns highest probability, i.e., favoring solutions "natural" to the student. However, we find that this approach works within a single teacher but fails when scaling to long reasoning traces from multiple diverse teachers. We identify a key cause: this approach scores entire solutions, but students generalize by recombining familiar reasoning steps, not by memorizing complete solutions. Full-trajectory scoring optimizes the wrong target; it rewards global fluency while the transferable signal lies in local step transitions. We propose Local Average Log Probability (LALP), which scores each reasoning step using only a small window of preceding context, measuring whether each step is justified by its immediate premises rather than whether the full response looks natural to the student. LALP enables two practical use cases: selecting the best teacher before fine-tuning and curating training data from diverse teacher pools. Across math, coding, and science reasoning tasks, LALP consistently improves accuracy when selecting the most natural solutions by a large margin.

## 1. Introduction

Training large language models (LLMs) to reliably perform multi-step reasoning has become a central challenge. Two complementary paradigms have emerged. Reinforcement learning from verifiable rewards can discover new reasoning behaviors by optimizing task-level outcomes (Lambert et al., 2025; Shao et al., 2024). Supervised distillation from stronger teacher models offers a more direct route to transferring reasoning ability into smaller, deployable students (Hinton et al., 2015). In practice, distillation can be surprisingly strong: for example, DeepSeek reports a distilled 7B student that surpasses RL-trained 32B models on reasoning benchmarks (DeepSeek-AI et al., 2025). These results make data curation for distillation a primary lever for improving reasoning capability under tight compute and deployment constraints.

Most existing curation methods focus on prompt selection, i.e., choosing questions to maximize diversity, difficulty, or coverage (Ash et al., 2020; Sorscher et al., 2022; Albalak et al., 2024; Yang et al., 2025b; Yu et al., 2025a). This line of work treats each prompt as having a single canonical target. Modern reasoning pipelines, however, often have access to multiple teachers (e.g., DeepSeek-R1, QwQ, Qwen3), and each teacher can produce multiple distinct, answer-correct solutions to the same prompt. These candidate responses can differ dramatically in verbosity, stylistic scaffolding, and step structure. This motivates the question we study: *given multiple candidate responses for the same prompt, which response should we train on?*

A natural approach is influence-based selection: estimate each candidate's downstream impact and train on the most influential traces. However, influence estimation (Xia et al., 2024) typically requires expensive gradient computations, making it impractical at the scale of long-form reasoning data. GRAPE (Zhang et al., 2025) proposes a simple alternative that is both efficient and principled: choose the response that the student model assigns highest probability and select the response that best fits the student's pretrained distribution. This probability-based criterion yields substantial gains in instruction-following benchmarks where responses are relatively short (Zhang et al., 2025).

We investigate whether this probability-based selection principle extends to long-form reasoning from heterogeneous teachers. Intuitively, it should: when candidates vary widely in form, selecting what is "natural" to the student seems like it ought to reduce training difficulty and improve outcomes. Interestingly, in the long, mixed-teacher regime, we find that global likelihood produces the wrong ranking: teachers whose traces receive higher probability under the student can yield lower downstream accuracy after fine-tuning. The

---

[1]Department of Electrical and Computer Engineering, Virginia Tech, Blacksburg, USA. Correspondence to: Hoang Anh Just <just@vt.edu>, Myeongseob Ko <myeongseob@vt.edu>, Ruoxi Jia <ruoxijia@vt.edu>.

*Proceedings of the 43rd International Conference on Machine Learning*, Seoul, South Korea. PMLR 306, 2026. Copyright 2026 by the author(s).

criterion that works for short, single-teacher responses fails for long, multi-teacher reasoning.

We hypothesize that the key reason is a granularity mismatch between what this likelihood measures and how reasoning generalizes. We analyze student generalization in representation space and find a consistent pattern: test solutions rarely resemble any single training trajectory, while individual reasoning steps are densely covered by the training data. This suggests that students generalize by recombining reusable steps rather than retrieving whole solutions. This suggests selection should evaluate step-level learnability, not trajectory-level fluency.

We therefore propose Local Average Log Probability (LALP), which scores responses at the granularity where reasoning appears to transfer: the step. LALP segments each candidate into reasoning steps, evaluates each step under a small local context window, and averages across steps. This measures whether each transition is locally supported by its immediate premises, rather than whether the full document reads smoothly under long-range conditioning.

LALP enables two practical forms of curation. First, for teacher selection: LALP scores computed before fine-tuning correctly predict which teacher's data yields the best student performance; global likelihood predicts the opposite. Second, for per-prompt response curation from a mixed-teacher pool, LALP improves math accuracy by up to 9.4% over global likelihood selection. These gains transfer to science reasoning and code generation suggesting that this local scoring rule is a broadly useful principle for reasoning data selection.

## 2. Related Work

The task of curating effective data for SFT of LLMs is an active and critical area of research. Our work builds upon and differentiates itself from several existing lines of inquiry, particularly in synthetic reasoning data generation, model-aware data selection, and knowledge distillation.

**Synthetic reasoning data.** The use of LLMs to generate synthetic chain-of-thought (CoT) responses (Wei et al., 2022) has become prevalent for tasks requiring multi-step inference (DeepSeek-AI et al., 2025; Ye et al., 2025; Liu et al., 2025; Bercovich et al., 2025). Recent work has also explored critique-and-revision pipelines using AI evaluators to filter or refine generated examples (Wu et al., 2025; Guha et al., 2026; Chen et al., 2025; Jiang et al., 2025). However, as Guha et al. (2026) and Chandra et al. (2025) observed, stronger teachers are not always more beneficial for a given student. Our work addresses the complementary question: when multiple teacher-generated responses are available, how should we select among them?

**Model-aware data selection.** Recognizing that one-size-fits-all SFT data is suboptimal (Li et al., 2025; Chandra et al., 2025), researchers have explored student-aware selection strategies. GRAPE (Zhang et al., 2025) made a significant step by selecting data based on the student's global average log probability of the entire response, aiming to choose sequences "natural" to the student's distribution. Our work directly builds on GRAPE, acknowledging its strengths for single-teacher, short responses but identifying a critical failure mode, the fluency trap, for mixed-teacher settings with long reasoning traces. Other approaches include offline surrogate modeling (Kostrikov et al., 2022; Bai et al., 2021), model gradient analysis (Jung et al., 2025; Panigrahi et al., 2025), online curriculum learning (Liang et al., 2021; Lu & Zhang, 2021), and active learning methods like SIFT (Hübotter et al., 2025), which combines retrieval and uncertainty reduction. Influence functions have also been explored (Choe et al., 2025; Humane et al., 2025), though they face scalability challenges for LLMs. Our method offers a simpler, more direct way to achieve model-awareness by operating on the student's inherent probabilities at the step level, without auxiliary models or complex training modifications.

**Knowledge distillation.** Our approach shares conceptual similarities with knowledge distillation (KD) (Hinton et al., 2015), where knowledge is transferred from a (typically larger) teacher to a smaller student. Classical KD aligns models at the token level by minimizing KL divergence between output distributions at every decoding step (Gou et al., 2021; Song et al., 2025), but this is computationally expensive. A more efficient alternative is response-level KD: the teacher generates complete responses, and the student is trained on those sequences with ordinary cross-entropy loss (Hsieh et al., 2023; Gupta et al., 2023; Ding et al., 2025). Our method belongs to this family but adds a student-aware step-level filter that scores local reasoning transitions rather than entire trajectories.

## 3. Problem Setting and Baseline

### 3.1. Response Selection for SFT

We study response selection for SFT. Given a prompt $x$ and a candidate set of responses $\mathcal{Y}(x) = \{y^{(1)}, \ldots, y^{(K)}\}$, generated by sampling multiple teacher models, the goal is to select a single response $y^* \in \mathcal{Y}(x)$ to include in the training set. Throughout the paper, all candidates' final answers match the ground truth. Selection therefore focuses not on correctness, but on supervision quality. Which reasoning trace, among several correct ones, provides the most effective training signal for a given student?

Formally, we seek a scoring function $f(y; x, \theta_S)$ that uses the student model (with parameters $\theta_S$) to evaluate candi-

dates, such that selecting $y^* = \arg\max_{y \in \mathcal{Y}(x)} f(y; x, \theta_S)$ yields the best downstream performance after fine-tuning on the selected data.

## 3.2. Baseline: Global Average Log Probability (GALP)

A recent approach to this problem is GRAPE (Zhang et al., 2025), which scores each candidate response by its global average log probability under the student model:

$$\mathrm{GALP}(y \mid x) = \frac{1}{m} \sum_{t=1}^{m} \log P_{\theta_S}(y_t \mid y_{<t}, x) \quad (1)$$

$$\propto \log P_{\theta_S}(y|x) \quad (2)$$

where $m$ is the number of tokens in $y$.

GRAPE's hypothesis is that SFT is most effective when training data aligns with the student's pretrained distribution. Responses that are globally "natural" to the student, those it assigns high probability, should be easier to learn and lead to better outcomes. This criterion is also computationally attractive: scoring requires only a single forward pass per candidate. In this paper, we stress-test GALP in a regime it was not originally tested on: long reasoning traces (over 12K tokens) drawn from heterogeneous teachers with systematically different verbosity and style.

## 4. When Global Selection Fails

This section presents the empirical motivation for our approach: in long-form reasoning distilled from heterogeneous teachers, selecting by GALP becomes an unreliable proxy for supervision quality. We begin with a controlled within-teacher sanity check where GALP behaves as intended, and then show two breakdowns in the long mixed-teacher regime: a teacher-level ranking reversal and degraded per-prompt selection.

**When GALP Works (Sanity Check).** We first verify that GALP is not a strawman. When candidate responses are generated by the same teacher (comparable style and similar lengths), selecting higher-GALP responses improves downstream accuracy. Table 1 shows results in this controlled setting. Within each teacher, we partition responses into the lowest, middle, and highest GALP terciles (computed by the pre-SFT student) and fine-tune the student on each subset. Selecting from the highest-GALP tercile consistently outperforms selecting from the lower terciles.

**Empirical Evidence for Failure.** We now consider long-form reasoning responses from three modern teachers: DeepSeek-R1, Qwen3-32B, and QwQ-32B. These models differ substantially in verbosity, self-reflection patterns, and stylistic scaffolding. For each teacher, we generate answer-verified solutions to the same prompts and fine-tune

| Student: Qwen2.5-7B-Instruct | | AVG |
|---|---|---|
| | Original Model | 0.353 |
| **Teacher:** | Lowest GALP | 0.292 |
| **Qwen2.5-72B** | Middle GALP | 0.338 |
| **-Instruct** | Highest GALP | 0.368 |
| **Teacher:** | Lowest GALP | 0.320 |
| **QwQ-32B** | Middle GALP | 0.349 |
| | Highest GALP | 0.382 |

*Table 1.* Within-teacher sanity check for GALP. For each teacher, responses are partitioned into terciles by the pre-SFT student's global average log probability (Eq. 1). Fine-tuning on the highest-GALP tercile yields the strongest average downstream performance.

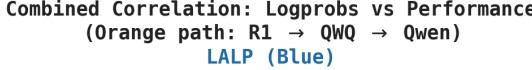
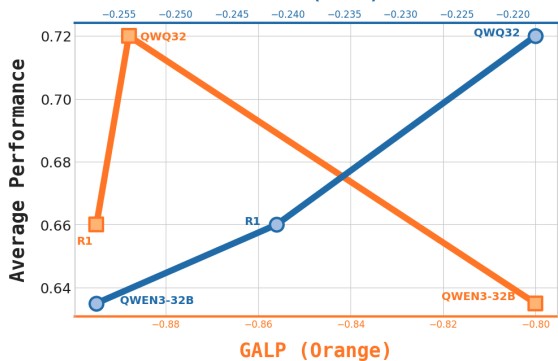

*Figure 1.* Cross-teacher reversal under global likelihood. For each teacher, we plot downstream accuracy after SFT on that teacher's dataset versus the pre-SFT student's average GALP on the same data (scaled by $10^2$ for readability). Across both students, higher average GALP does not imply higher downstream accuracy.

the same student on each teacher-specific dataset. Figure 1 shows that ranking teachers by their average GALP under the student does not predict and can even invert the ranking by downstream accuracy. Specifically, Qwen3-32B traces receive the highest average GALP under both 7B and 32B students, yet SFT on these traces yields the worst downstream accuracy; QwQ-32B traces do not achieve the highest GALP, yet SFT on them yields the best downstream accuracy.

We further test GALP in a more fine-grained setting: per-prompt selection from a mixed-teacher pool. For each prompt, we pool answer-verified responses from all three teachers and select one response per prompt using: random selection and highest GALP. We then fine-tune the same student on each curated dataset. As shown in Table 3, in this mixed-teacher setting, GALP selection underperforms random selection for the 32B student and provides only marginal gains for the 7B student.

## 5. Why GALP Fails

GALP scores a candidate solution by the probability of the full response given the prompt, implicitly treating trajectory likelihood as a proxy for supervision quality. In long-form reasoning, we argue this target is misaligned with how students actually generalize. We provide evidence that reasoning transfers at the granularity of steps, not trajectories, and draw implications for how selection should be designed.

**Hypothesis: Step-compositional Generalization.** We probe the granularity at which reasoning supervision transfers from training to test time. For each held-out test solution, we embed it at two levels: (i) a single embedding for the full trajectory, and (ii) separate embeddings for individual reasoning steps. We quantify coverage as the cosine similarity to the nearest training-set neighbor at each level. (See Appendix B for step extraction details.) Higher similarity indicates that test-time reasoning is well-supported by training examples at that granularity.

Figure 2 shows an interesting gap: step-level embeddings have very high nearest-neighbor similarity (0.935), while full-trajectory embeddings are substantially less covered (0.760). This indicates that, at test time, models rarely encounter a training example that matches an entire solution template, but they likely encounter locally similar reasoning moves. Figure 2 confirms this gap is consistent across test problems. We observe a similar trend across benchmarks we have tested and provide further results in Appendix C.6.

These statistics support a step-compositional view of reasoning: students improve by learning a library of reusable local transitions and recombining them into novel trajectories. This perspective aligns with the basic structure of autoregressive inference: at each point in a solution, the model must produce a plausible next step given the prompt and recent progress. If what transfers is the step, then the learnability of a candidate solution should be governed by the quality of its local transitions, not by the likelihood of the entire response under an arbitrarily long prefix that is largely problem-specific.

**Implication for Likelihood-Based Selection.** The coverage gap implies that optimizing for high-probability trajectories is a mismatched target for selection in this regime: full trajectories are novel by design, while their constituent steps are what recur across problems. Selection should instead prioritize solutions whose individual steps are locally natural and reusable for the student. We formalize this intuition in a toy theoretical model (Appendix A), which shows that when a learner has dense support only for local contexts, it acquires signal there but collapses to an uninformative prior for non-local contexts, making global scoring unreliable while local scoring remains informative.

## 6. Method: Local Average Log Probability

The analysis above suggests that likelihood-based selection should (i) condition on the level where reasoning generalizes, steps rather than full context, and (ii) aggregate at the step level. We formalize this as **Local Average Log Probability (LALP)**.

**Step segmentation.** LALP requires partitioning a response into a sequence of reasoning steps. We use a capable, open-weight large language model, GLM-4.5-Air (Zeng et al., 2025a), which can handle large context that is necessary for our long reasoning data. Full implementational provided in Appendix C.

**Definition.** Let a response $y$ be segmented into $p$ steps, $y = (s_1, s_2, \ldots, s_p)$, where each step $s_i = (s_i^1, s_i^2, \ldots, s_i^{|s_i|})$ is a sequence of $|s_i|$ tokens. We score each step by its token-average log probability under the student model $\theta_S$, conditioning only on the prompt $x$ and the $k$ immediately preceding steps:

$$\text{LocalLP}(s_i) = \frac{1}{|s_i|} \sum_{t=1}^{|s_i|} \log P_{\theta_S}\big(s_i^t \mid s_i^{<t}, \\ s_{\max(1, i-k):i-1}, x\big).$$

where $s_i^{<t}$ denotes the tokens within step $i$ before position $t$. LALP is the average of these step scores:

$$\text{LALP}(y \mid x) = \frac{1}{p} \sum_{i=1}^{p} \text{LocalLP}(s_i). \tag{3}$$

Given multiple candidate responses for a prompt, we select the response with the highest $\text{LALP}(y \mid x)$.

This construction enforces two properties that GALP lacks in the long mixed-teacher regime: (i) equal step weighting (each reasoning move contributes equally, independent of verbosity), and (ii) local conditioning (each step is evaluated for standalone plausibility given its immediate premises, rather than for global document-level fluency).

**Context window size.** The window size $k$ governs the degree of locality. To study its effect, we set the window size to a fraction $\alpha \in \{5\%, 25\%, 50\%, 75\%\}$ of the available prefix steps for each response, and compare the resulting teacher rankings to the GALP ranking. Figure 3 shows a clear transition: with small windows ($\alpha = 5\%$–$25\%$), rankings are stable and match downstream performance (correctly identifying QwQ-32B as the strongest teacher). As $\alpha$ increases, rankings converge toward GALP, eventually

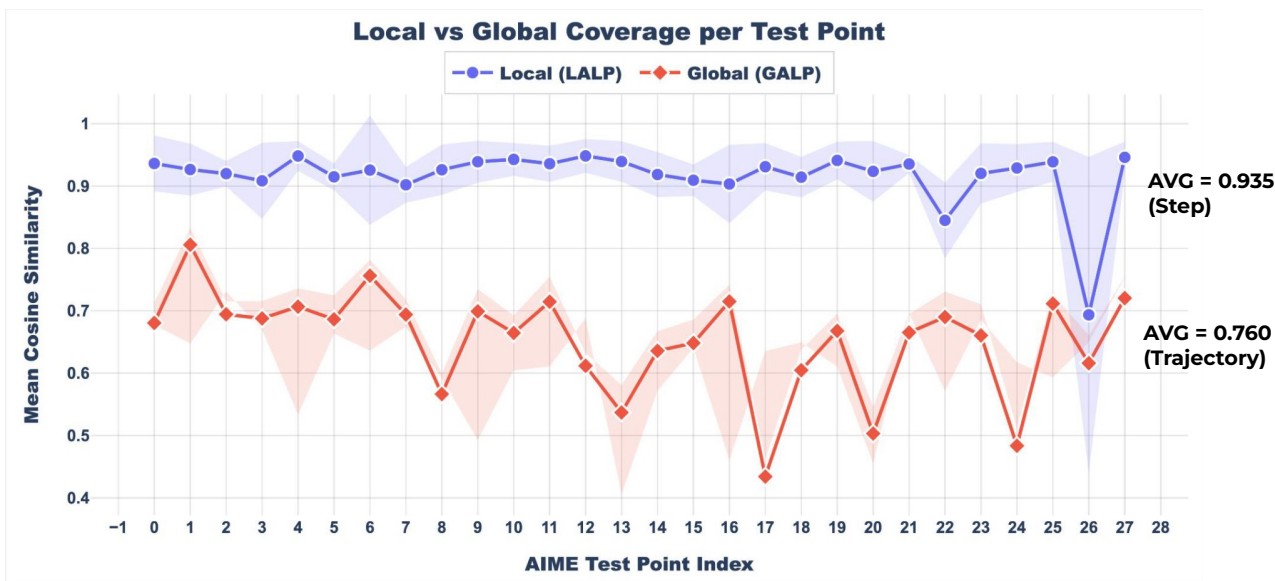

*Figure 2.* Per-test-point coverage distributions for AIME 2025 problems. Each violin shows the distribution of cosine similarities between a test point and training data. **Red (Global):** Trajectory-level embeddings show sparse coverage with high variance across test points (mean=0.760, range 0.68–0.83). Many test problems have no close training match. **Blue (Local):** Step-level embeddings exhibit dense, consistent coverage (mean=0.935, p10=0.89) with tight distributions per test point, allowing reasoning step to find a similar training step. This 23% coverage gap confirms that step-level representations generalize reliably while full trajectories do not.

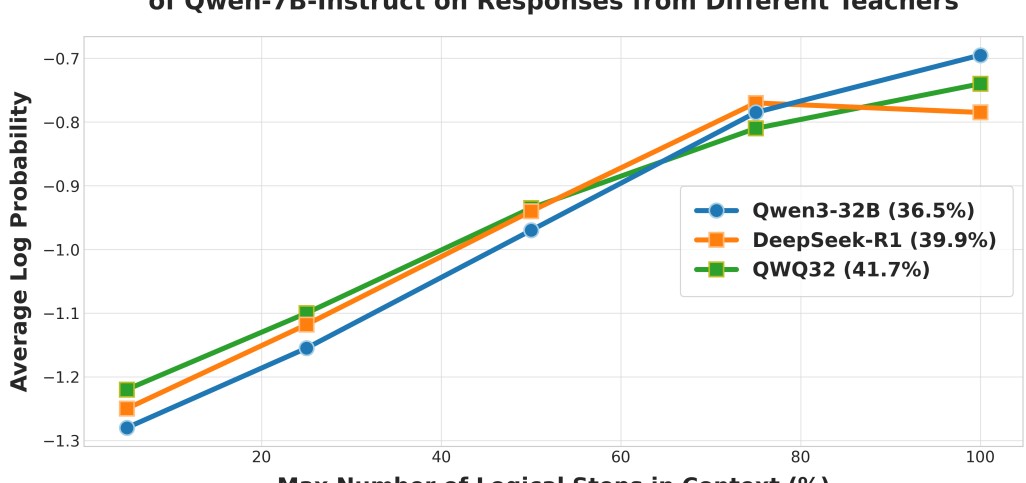

*Figure 3.* **Effect of context window on teacher ranking.** Mean step-level log probability (LALP) computed with increasing context windows. Small windows (5–25%) yield a teacher ordering that matches downstream performance; large windows (50–75%) converge to the GALP ordering and recover the cross-teacher reversal.

recovering the same failure mode where Qwen3-32B ranks highest despite yielding the weakest fine-tuned accuracy.

This ablation indicates that locality is the driver of LALP's advantage and that rankings are robust across a range of small windows.

**Practical considerations.** Computing LALP requires scoring each step under its local window, which entails $p$ forward passes per response (vs. a single pass for GALP). In practice, this overhead is manageable for two reasons. First, selection is a one-time preprocessing cost incurred before fine-tuning. Second, for teacher selection, we find that scores computed on as few as 200 prompts reliably recover the full-dataset teacher ranking (Section 7.1). Finally, steps

from different responses can be batched to improve GPU utilization.

| Model Configuration | OLYMPIAD | CN_MATH24 | AVG |
|---|---|---|---|
| **Student: Qwen2.5-7B-Instruct** | | | |
| Original Model | 0.404 | 0.167 | 0.353 |
| Random | 0.456 | 0.233 | 0.407 |
| GALP | 0.441 | 0.233 | 0.412 |
| Local Lowest | 0.433 | 0.233 | 0.399 |
| LALP | 0.441 | 0.333 | **0.440** |
| **Student: Qwen2.5-32B-Instruct** | | | |
| Original Model | 0.471 | 0.233 | 0.445 |
| Random | 0.636 | 0.733 | 0.651 |
| GALP | 0.636 | 0.733 | 0.632 |
| Local Lowest | 0.640 | 0.700 | 0.623 |
| LALP | 0.673 | 0.833 | **0.726** |

*Table 2.* Teacher selection results showing performance on OlympiadBench, Chinese Math, and the Average over 7 benchmarks. Full results in Table 9.

# 7. Evaluation

In this section, we evaluate LALP as a data-selection rule for reasoning distillation in the long, mixed-teacher regime. Our experiments address two practical use cases: (1) selecting the best teacher model before fine-tuning, and (2) per-prompt response selection from a mixed-teacher pool. We find that LALP correctly ranks teachers and improves downstream accuracy by up to 9.4% over GALP in per-prompt selection. We then analyze three mechanisms underlying these gains: shifted attribution toward reasoning-critical tokens, resistance to self-conditioning artifacts, and a dissociation between training loss and generalization that supports the step-compositional view of reasoning.

**Experimental setup.** We use LIMO prompts (Ye et al., 2025) for long-form mathematical reasoning. To test cross-domain transfer, we additionally evaluate on GPQA-Diamond (Rein et al., 2024) (science reasoning) and Live-CodeBench (Jain et al., 2025) (code reasoning). We consider three student models: Qwen2.5-7B-Instruct, Qwen2.5-32B-Instruct, and Llama-3.1-8B-Instruct. Our teacher pool contains DeepSeek-R1 (DeepSeek-AI et al., 2025), Qwen3-32B-Instruct, and QwQ-32B, which differ substantially in verbosity and stylistic conventions. We report average accuracy across a diverse suite of math benchmarks (MATH, AIME 2025, AMC, MINERVA, KAOYAN, Olympiad-Bench, CN_MATH24); full evaluation details and hyperparameters are provided in Appendix B.

## 7.1. Teacher Model Selection

We first test a coarse-grained but practical use case: selecting a teacher before fine-tuning. Fine-tuning a student on each candidate teacher is expensive; an effective scoring rule

should predict the best teacher using only the pre-trained student.

For each teacher, we compute the average GALP and LALP of its responses under the pre-trained student, then fine-tune the student on that teacher's dataset and measure downstream accuracy. Table 2 shows that LALP correctly ranks teachers by downstream utility (QwQ > DeepSeek-R1 > Qwen3) for both 7B and 32B students, while GALP induces the opposite ordering. In addition, LALP rankings computed from as few as 200 prompts match those from the full 817-prompt set, enabling efficient teacher selection in practice.

## 7.2. Per-Prompt Response Selection from a Mixed-Teacher Pool

| Model Configuration | AVG |
|---|---|
| **Student: Qwen2.5-7B-Instruct** | |
| Original Model | 0.353 |
| Random | 0.407 |
| GALP | 0.412 |
| Local Lowest | 0.399 |
| LALP | **0.440** |
| **Student: Qwen2.5-32B-Instruct** | |
| Original Model | 0.445 |
| Random | 0.651 |
| GALP | 0.632 |
| Local Lowest | 0.623 |
| LALP | **0.726** |

*Table 3.* Response selection from mixed-teacher pool. LALP (Local Highest) outperforms GALP (Global Highest) by **+9.4%** on the 32B student (0.726 vs 0.632). Notably, LALP-selected data even outperforms training on all responses from the best single teacher (QwQ-32B: 0.719 in Table 2). Full results in Table 10.

Teacher selection treats each source as monolithic. In modern distillation pipelines, practitioners often have multiple candidate traces per prompt (multiple teachers, or multiple samples). We therefore consider the fine-grained setting: for each prompt, pool correct traces from all teachers and select a single response.

We compare four selection rules: (1) Random, (2) GALP (highest global log probability), (3) LALP (highest local log probability), and (4) Local Lowest (sanity check). Students are fine-tuned on the resulting 817-example curated datasets.

Table 3 shows that LALP yields the best downstream accuracy for both students, improving by **+9.4** points over GALP on Qwen2.5-32B-Instruct (0.726 vs. 0.632). Notably, GALP underperforms random selection for the 32B student, indicating that in this regime global likelihood is not merely noisy but systematically misranked.

Finally, the LALP-selected dataset, a mixture drawn from

all three teachers, outperforms training on the single best teacher alone (QwQ-32B; Table 2). This suggests that step-aware selection can exploit complementary strengths across teachers without committing to one source.

### 7.3. Generalization Beyond Math

To test whether LALP's selection principle extends beyond mathematics, we evaluate models trained on LIMO-selected math data on science (GPQA-Diamond) and code (Live-CodeBench). Table 16 shows that LALP improves over GALP on both benchmarks (+9.1 points on GPQA-Diamond and +2.9 points on LiveCodeBench-Hard), suggesting the gains are not specific to math-only evaluation.

### 7.4. Mechanisms Behind LALP's Gains

The step-transferability hypothesis (Section 5) explains why trajectory-level scoring is misaligned with reasoning generalization. Here we trace LALP's improvements to three concrete mechanisms that follow from its design.

#### 7.4.1. TOKEN-LEVEL ATTRIBUTION.

Long solutions contain two broad classes of tokens:

- **Reasoning-critical tokens**: calculations, bindings, and logical transitions that introduce specific new information and are often locally low-probability.

- **Discourse tokens**: scaffolding (e.g., "Okay," "Therefore," restatements) that is abundant and highly predictable.

Because GALP averages over all tokens uniformly, abundant discourse tokens dominate the score by sheer volume, while sparse reasoning-critical tokens contribute minimally. This makes GALP more sensitive to stylistic fluency than to reasoning quality, a phenomenon we formalize as the "fluency trap" in Theorem A.3 (Appendix A), which shows that when high-probability style tokens outnumber low-probability reasoning tokens, GALP can systematically prefer incorrect responses. We quantify this effect by categorizing tokens in 500 long responses and measuring each method's attribution mass per category. In particular, to measure where each method focuses its "attention," we compute the *probability mass* attributed to each token category. Specifically, for each token $t$ with log probability $\log p_t$, we convert to probability $p_t = \exp(\log p_t)$ and compute the fraction of total probability mass in each category: $\text{mass}_c = \sum_{t \in \text{category } c} p_t / \sum_t p_t$. High-probability tokens (e.g., discourse fillers) naturally dominate GALP's mass because they contribute disproportionately to the sum. For LALP, the mass is computed analogously using local probabilities conditioned on each split's prefix. As shown in Table 4, GALP assigns 42% of its mass to discourse tokens, while math/symbols and reasoning transitions together receive only 44%. LALP

| Token category | GALP (mass %) | LALP (mass %) |
|---|---|---|
| Discourse / filler (e.g., "Okay", "So") | 42.3 | 18.7 |
| Math / symbols (e.g., variables, operators) | 31.2 | 48.6 |
| Reasoning transitions (e.g., "therefore") | 12.8 | 22.4 |
| Other | 13.7 | 10.3 |

*Table 4.* Token-level signal analysis via attribution mass. Under GALP, a large fraction of weight falls on predictable discourse tokens, while LALP shifts emphasis toward math/symbol tokens and reasoning transitions.

reverses this balance: discourse tokens account for just 19%, while math and reasoning tokens together receive 71%—a shift of 27 percentage points toward reasoning-bearing content. Figure 4 reveals a complementary pattern among low-probability tokens, which are most informative for distinguishing candidates under likelihood-based scoring. LALP surfaces substantially more math/symbol tokens in this critical region than GALP, indicating that step-level aggregation preserves signal from the sparse logical transitions that token-averaging dilutes. Theorem A.4 formalizes this robustness: LALP's separation between correct and incorrect responses dilutes with the number of *steps* $p$, not the number of *tokens* $m$, providing greater resilience to verbose scaffolding.

#### 7.4.2. LOCAL CONDITIONING RESISTS SELF-REINFORCEMENT.

Full-context scoring in GALP introduces a subtler problem: as context accumulates, later tokens become increasingly predictable simply by being consistent with earlier text—regardless of reasoning quality. Figure 5 quantifies this self-reinforcement effect across all three teachers. Panel (a) shows that the gap between global and local log probability grows steadily with position: by the end of a response, global scoring assigns 0.8–1.1 nats more probability than local scoring, purely from accumulated context. Panel (b) reveals the underlying mechanism: global LP (dashed lines) rises throughout responses as the model becomes increasingly confident from self-referential context, while local LP (solid lines) remains flat, reflecting only the intrinsic difficulty of each step. This self-conditioning explains why verbose, repetitive responses score highly under GALP: once the model commits to a pattern or intermediate value, restating it is nearly free. Qualitatively, we observe GALP-preferred responses frequently repeat earlier steps or values, and the model becomes increasingly confident each time (with examples below). LALP, by conditioning only on local context, scores each step for its intrinsic plausibility rather than its consistency with accumulated text.

**Qualitative Evidence: Self-Conditioning in Action** Before presenting aggregate statistics, we illustrate self-conditioning through concrete examples from actual model generations. These examples demonstrate how global log

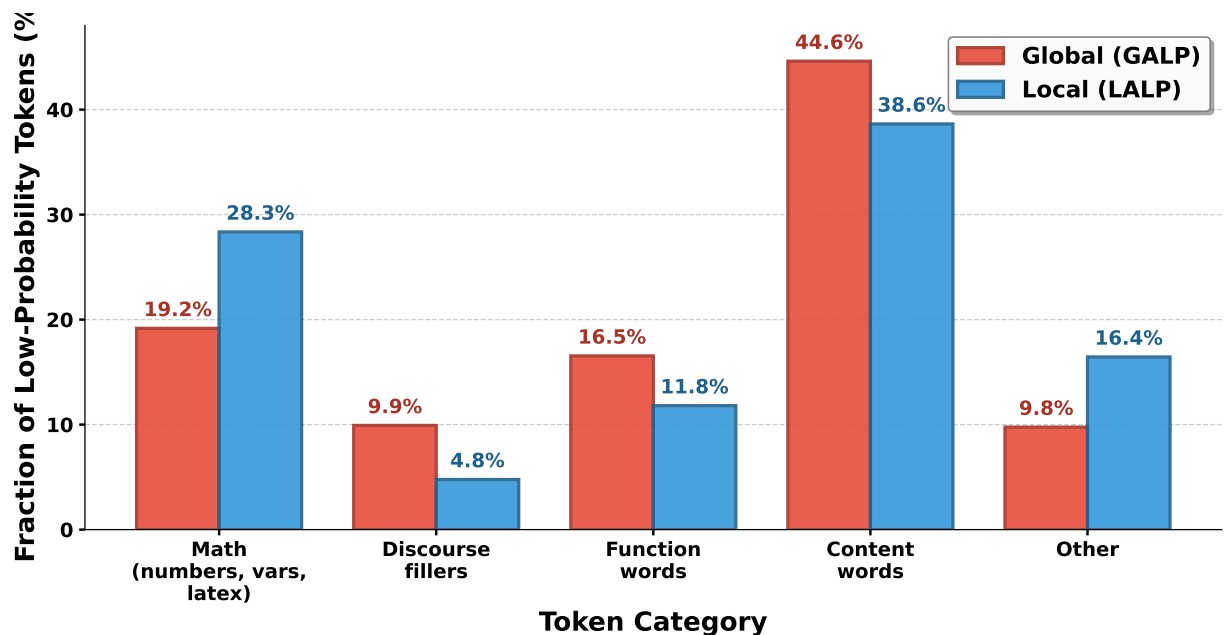

*Figure 4.* Token type distribution among low-probability tokens under GALP vs. LALP. Local scoring identifies a larger fraction of math/symbol tokens among low-probability regions, indicating better focus on reasoning-bearing content. Analysis over 440 LIMO responses across three teachers.

probability inflates confidence for tokens that merely echo earlier content. We provide additional examples in Appendix due to space constraints.

**Example 1: Numerical self-conditioning ("177").** Consider a response where the model calculates the final answer $m + n = 177$ three times. We track the probability of the last digit "7" across occurrences:

| Occurrence | Log prob of "7" | Probability |
|---|---|---|
| First calculation | $-1.83$ | 16.0% |
| Verification step | $-0.10$ | 90.4% |
| Final answer | $-0.00003$ | 99.99% |

The probability jumps from 16% to 99.99% not because the model became more certain of the *correctness* of the calculation, but because it is now copying from its own context. The initial 16% probability reveals genuine computational uncertainty; the subsequent near-certainty reflects only self-conditioning.

**Example 2: A toy demonstration.** To isolate the phenomenon, consider the deliberately incorrect statement: "1+1 is 3, so the answer is 3":

| Token | Log probability | Probability |
|---|---|---|
| "3" (first occurrence) | $-3.01$ | 4.9% |
| "3" (second occurrence) | $-0.02$ | 97.7% |

The model initially assigns low probability to the incorrect

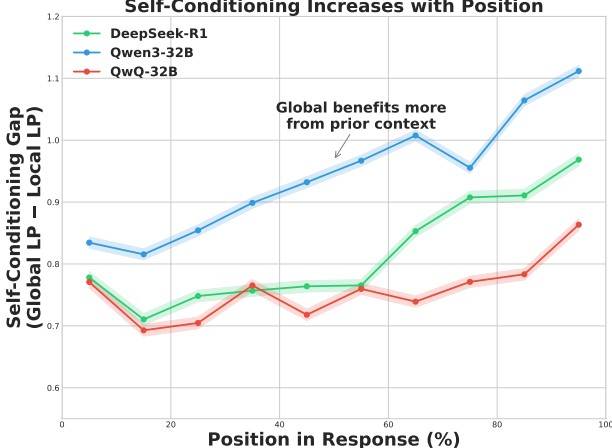

*Figure 5.* **Self-conditioning increases with position.** (a) The gap between global and local log probability grows monotonically across all three teachers, confirming that global scoring increasingly rewards tokens that echo earlier content. (b) Global LP rises with position (self-conditioning effect) while local LP remains relatively flat, demonstrating that LALP isolates reasoning difficulty from contextual repetition.

answer "3" (4.9%), reflecting appropriate uncertainty. But after committing to this error, it assigns 97.7% probability to repeating "3" as the final answer. Under GALP, this self-confirmed error receives a high average score, while LALP by truncating context can detect that the second "3" lacks local justification.

**Implication.** These examples reveal that global scoring conflates two distinct signals: (i) *reasoning confidence* (whether a step is logically justified) and (ii) *recall confidence* (whether a token matches earlier context). LALP separates these by evaluating each step with limited context, preventing prior commitments from inflating scores. We now quantify this effect systematically.

**Quantitative Evidence: Self-Conditioning Gap vs Position** The qualitative examples above suggest that self-conditioning should increase with position in the response. Early tokens have little prior context to echo, while later tokens can leverage extensive earlier text. We test this prediction by measuring the *self-conditioning gap*, the difference between global and local log probability, as a function of position across all responses.

**Setup.** For each token at position $t$ in a response, we compute:

$$\begin{aligned}
\text{Gap}(t) = &\log P_{\text{global}}(y_t \mid y_{<t}, x) \\
&- \log P_{\text{local}}(y_t \mid y_{t-k:t-1}, x).
\end{aligned} \quad (4)$$

where $k$ is the local context window size. Positive gap indicates that global scoring assigns higher probability than local scoring, indicating self-conditioning. We bin tokens by their relative position (0-100% of response length) and compute mean gap per bin across ∼800 responses from each of three teacher models (DeepSeek-R1, QwQ-32B, Qwen3-32B).

**Result.** Figure 5 confirms the prediction: the self-conditioning gap increases monotonically with position for all three teachers. At the beginning of responses (0-10%), the gap is around 0.7-0.85. By the end (90-100%), the gap increases by at least 0.1 to almost 0.3 across teachers, indicating that global scoring inflates probabilities for tokens late in responses.

**Interpretation.** The monotonic increase in self-conditioning gap has two important implications:

1. **GALP overweights late tokens:** Later tokens contribute disproportionately high log probabilities to the global average, even when they merely repeat earlier content. This biases selection toward responses with more repetition in their latter portions.
2. **LALP normalizes across position:** By truncating context, local scoring assigns similar probabilities to equivalent reasoning moves regardless of position, properly measuring step difficulty rather than recall ease.

Qwen3 shows the largest self-conditioning gap at all positions, consistent with its verbose, repetitive style. This explains why Qwen3 achieves the highest GALP scores

(Table 2) despite producing the worst downstream student performance, its high global scores are inflated by self-conditioning rather than reflecting quality reasoning.

### 7.4.3. SLOWER LEARNING YET BETTER GENERALIZATION.

Figure 6 shows that GALP-selected data achieves lower training loss than LALP-selected data, yet yields worse downstream accuracy (Table 3). One interpretation is that the student fits globally fluent scaffolding easily, while the sparse reasoning transitions contribute minimally to the loss. Consistent with this view, LALP-selected data exhibits the opposite pattern: higher global training loss, but lower local NLL on held-out reasoning steps, suggesting the student has learned more reusable step-level patterns. These findings support the hypothesis that reasoning performance is less dependent on trajectory fluency and more on the learnability of individual steps.

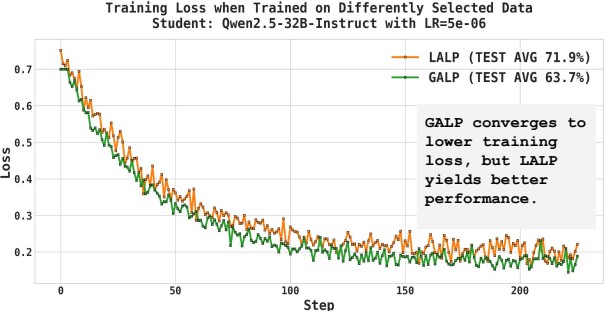

*Figure 6.* Training loss when fine-tuning on differently selected data (student: Qwen2.5-32B-Instruct). Global selection achieves lower training loss, but local selection yields higher downstream reasoning accuracy, illustrating a loss–generalization mismatch.

## 8. Conclusion

We identify a fundamental mismatch between how reasoning models generalize and how existing likelihood-based methods score candidate training data. While global average log probability (GALP) works well for short, single-teacher responses, it fails in the increasingly common regime of long reasoning traces from diverse teachers. The key issue is granularity: GALP scores entire trajectories, but students generalize by recombining familiar reasoning steps into novel solutions. Full-trajectory scoring rewards global fluency while the transferable signal lies in local step transitions. To fix this, we propose Local Average Log Probability (LALP). By scoring transitions within a small local window, LALP focuses on whether steps are justified by their immediate premises. This shift aligns data selection with how models actually learn, yielding up to a 9.4% accuracy gain in math, better teacher selection, and strong cross-domain transfer to science and code.

**Acknowledgement.** Ruoxi Jia and the ReDS lab acknowledge support through grants from the National Science Foundation under grants IIS-2312794, IIS-2313130, and OAC-2239622.

## Impact Statement

This paper advances the field of Machine Learning by exploring the intersection of data structuring and model generalization. While we do not identify specific immediate societal risks, the methodologies proposed, specifically logical segmentation and dynamic data selection, offer a path toward more interpretable multi-step inference systems. Future research may extend these principles beyond reasoning tasks to broader transfer learning settings, potentially improving how models use structured information in complex, real-world domains.

**Limitations.** LALP cannot verify long-range consistency, if an early step contains an error, later steps that correctly follow from it may still score highly. In settings where such consistency is critical, hybrid approaches combining local scoring with global coherence checks may be necessary.

Furthermore, our experiments focus on response selection for supervised fine-tuning (SFT). We do not evaluate LALP inside reinforcement learning (RL), although RL is an important part of modern reasoning model training pipelines. Since LALP provides a student-aware score for individual reasoning steps, it could potentially be used beyond SFT as a lightweight step-level signal. For example, as an auxiliary dense reward or as a way to curate step-level data for training process reward models. However, using LALP in RL introduces additional challenges that are outside the scope of this paper. LALP measures whether a step is locally learnable under the student model, not whether the step is objectively correct. Directly optimizing this signal could therefore reinforce plausible but incorrect reasoning or over-align training to the student's prior distribution, creating reward hacking. Extending LALP to RL would likely require careful designing.

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

## A. Toy theoretical model: locality induces a reasoning gap

Inspired by the locality-of-experience account of step-by-step reasoning (Prystawski et al., 2023), this appendix provides a simple theoretical model supporting the main intuition of our paper: if training observations provide dense support for *local* reasoning transitions but sparse support for whole trajectories, then a learner trained with cross-entropy plus smoothing/regularization can learn local transitions while reverting to an uninformative prior on under-supported long-range contexts. This induces a "reasoning gap" that makes whole-trajectory scoring unreliable and motivates step-level scoring such as LALP.

**Motivation.** This appendix provides a simple theoretical model supporting the main intuition of the paper, inspired by the locality-of-experience account of step-by-step reasoning (Prystawski et al., 2023). If training observations provide dense support for *local* reasoning transitions but sparse support for whole trajectories, then a learner trained with cross-entropy (plus smoothing/regularization) can learn local transitions while reverting to an uninformative prior on under-supported long-range contexts. This induces a "reasoning gap" that makes whole-trajectory scoring unreliable and motivates step-level scoring such as LALP.

While prior work studies locality over variables in a probabilistic graphical model, we use the same principle as an abstraction for reasoning data. Local reasoning patterns recur across problems, whereas complete solution trajectories are rarely repeated. Thus, in our setting, "local observation" should be understood as a proxy for **coverage/support density**, not as a literal claim that language models only observe adjacent reasoning steps.

**Connection to empirical observations.** Before presenting the formal model, we clarify how it connects to our main empirical findings. In Section 5, we observed that *step-level embeddings* are densely covered by training data (cosine similarity 0.935), while *trajectory-level embeddings* are sparsely covered (0.760). This coverage gap is the empirical manifestation of what we formalize below: contexts with substantial repeated support in training (local/step-like) allow the learner to acquire signal, while contexts that are effectively unseen (global/trajectory-like) force the learner to default to a prior.

The toy model below abstracts this as "local observation," which should be understood as a proxy for **coverage/support density** rather than a literal claim that language models only see adjacent tokens. The key insight is that *local reasoning patterns recur heavily across problems*, providing dense training signal, while *specific trajectory configurations are rare*, leaving the learner underspecified at that granularity.

**Setup (directed chain of steps).** Let $Y_1, \ldots, Y_N$ be discrete random variables taking values in a finite set $\mathcal{X}$ and assume the data distribution factorizes as a directed chain:

$$p_d(Y_1, \ldots, Y_N) = p_d(Y_1) \prod_{j=1}^{N-1} p_d(Y_{j+1} \mid Y_j). \tag{5}$$

We interpret $Y_j$ as an *atomic reasoning step*. The key assumption is **local observation**: training samples only expose local neighborhoods. Concretely, we observe pairs $(i, i+1)$ uniformly at random (or more generally, pairs within a window of size $k$), and we train a conditional model $q$ to predict a value given identifiers/previous observations.

**Risk with smoothing.** To model finite capacity, regularization, or inductive bias toward high-entropy predictors, consider the risk

$$R(q) = H(p, q) + \beta \, H(u, q), \tag{6}$$

where $H(\cdot, \cdot)$ is cross-entropy, $u$ is a uniform "background" distribution over the same support, and $\beta > 0$ controls the strength of smoothing.[1]

**Concrete distributions and model class.** Let $\mathcal{I} = \{1, \ldots, N\}$. We model a conditional predictor $q(\cdot \mid i, j, y)$ that outputs a distribution over $y' \in \mathcal{X}$ given an index pair $(i, j) \in \mathcal{I}^2$ and an observed value $y \in \mathcal{X}$ for $Y_i$. (This is the minimal abstraction needed for the locality argument.)

---

[1]This form is equivalent to a cross-entropy term plus an entropy-regularization prior; it is also a standard way to model a learner that interpolates between empirical conditionals and a background distribution.

Let $p_{\text{obs}}$ be uniform over adjacent pairs: $p_{\text{obs}}(i,j) > 0$ iff $j = i + 1$. Define the *local* data distribution $p$ over $(i, j, y, y')$ by

$$p(i, j, y, y') = \begin{cases} p_{\text{obs}}(i,j)\, p_d(Y_i = y, Y_{i+1} = y'), & j = i + 1, \\ 0, & \text{otherwise.} \end{cases}$$

Let $u$ be a uniform background distribution over the same variables (so in particular $u(y' \mid i, j, y) = 1/|\mathcal{X}|$ for all contexts).

Finally, define cross-entropy for discrete distributions as $H(p, q) = \mathbb{E}_{X \sim p}[-\log q(X)] = -\sum_x p(x) \log q(x)$.

The following elementary mixture-minimizer result is standard. We state it explicitly because it is the basic algebraic step used in the locality-based reasoning-gap analysis of Prystawski et al. (2023).

**Proposition A.1** (Mixture minimizer). *Let $p_1$ and $p_2$ be discrete distributions and $\alpha_1, \alpha_2 \geq 0$. The minimizer of $\alpha_1 H(p_1, q) + \alpha_2 H(p_2, q)$ over distributions $q$ is*

$$q^\star = \frac{\alpha_1}{\alpha_1 + \alpha_2} p_1 + \frac{\alpha_2}{\alpha_1 + \alpha_2} p_2.$$

*Proof.* Assume $p_1$ and $p_2$ have finite support $\Omega$ and $\alpha_1 + \alpha_2 > 0$.[2] Write the objective as

$$J(q) = \alpha_1 H(p_1, q) + \alpha_2 H(p_2, q) = -\sum_{x \in \Omega} \big(\alpha_1 p_1(x) + \alpha_2 p_2(x)\big) \log q(x),$$

subject to $q(x) \geq 0$ and $\sum_x q(x) = 1$.

Form the Lagrangian

$$\mathcal{L}(q, \lambda) = -\sum_{x \in \Omega} \big(\alpha_1 p_1(x) + \alpha_2 p_2(x)\big) \log q(x) \;+\; \lambda\Big(\sum_{x \in \Omega} q(x) - 1\Big).$$

For any $x$ with $q(x) > 0$, the first-order condition is

$$\frac{\partial \mathcal{L}}{\partial q(x)} = -\frac{\alpha_1 p_1(x) + \alpha_2 p_2(x)}{q(x)} + \lambda = 0,$$

so

$$q(x) = \frac{\alpha_1 p_1(x) + \alpha_2 p_2(x)}{\lambda}.$$

Enforcing normalization yields

$$1 = \sum_{x \in \Omega} q(x) = \frac{1}{\lambda} \sum_{x \in \Omega} \big(\alpha_1 p_1(x) + \alpha_2 p_2(x)\big) = \frac{\alpha_1 + \alpha_2}{\lambda},$$

so $\lambda = \alpha_1 + \alpha_2$. Substituting back gives

$$q^\star(x) = \frac{\alpha_1 p_1(x) + \alpha_2 p_2(x)}{\alpha_1 + \alpha_2},$$

which is a valid distribution. Since $J(q)$ is strictly convex in $q$ on the probability simplex whenever $\alpha_1 + \alpha_2 > 0$, this minimizer is unique. $\qquad\square$

**Relation to prior locality theory.** The following theorem adapts the reasoning-gap argument of Prystawski et al. (2023) from variables in a chain-structured probabilistic model to step-level contexts in reasoning traces. The proof follows the same high-level structure, where local contexts receive data support and therefore learn a smoothed version of the true transition, while unsupported non-local contexts collapse to a background distribution. Our use differs in interpretation, where "locality" denotes coverage/support density of reusable reasoning steps, rather than literal co-occurrence of variables in a Bayes net.

---

[2]If $\alpha_1 = \alpha_2 = 0$, then the objective is identically zero and any $q$ is a minimizer.

**Theorem A.2** (Reasoning gap under locality). *Assume the observation process provides dense support only for adjacent pairs $(i, i+1)$ (or more generally, pairs within a window of size $k$), while non-local pairs have negligible support. Let $q^\star$ minimize* (6). *Then:*

- *For adjacent steps $(Y_i, Y_{i+1})$, the learned conditional interpolates between the true transition and the background:*

$$q^\star(\cdot \mid i, i+1, y) = \lambda_{i,y}\, p_d(\cdot \mid Y_i = y) + (1 - \lambda_{i,y})\, u(\cdot) \quad \text{for some } \lambda_{i,y} \in (0,1).$$

- *For non-adjacent steps $(Y_i, Y_j)$ with $|i - j| > 1$, the model reverts to the background:*

$$q^\star(\cdot \mid i, j, y) = u(\cdot).$$

**Interpretation.** This result formalizes a general principle: *where data provides dense support, the learner acquires signal; where support is sparse, the learner collapses to a prior*. In our empirical setting, "local" corresponds to step-level contexts that recur across problems (dense manifold coverage), while "non-local" corresponds to specific trajectory configurations that are rarely seen (sparse coverage). The theorem explains why GALP can be unreliable: it aggregates predictions over contexts where the model has no informative signal and defaults to a background distribution. LALP, by evaluating within local windows where support is dense, queries the model only where it has learned meaningful transitions.

*Proof.* Following the cross-entropy decomposition used by Prystawski et al. (2023), we expand the risk (6) by conditioning on contexts. Write $c = (i, j, y)$ for a context and let $p(c)$ denote the marginal probability of $c$ under $p$. Then

$$
\begin{aligned}
H(p, q) &= \mathbb{E}_{(c,y') \sim p}\big[ -\log q(y' \mid c) \big] \\
&= \sum_c p(c)\, \mathbb{E}_{y' \sim p(\cdot \mid c)}\big[ -\log q(y' \mid c) \big] \\
&= \sum_c p(c)\, H\big(p(\cdot \mid c),\, q(\cdot \mid c)\big).
\end{aligned}
$$

Similarly,

$$H(u, q) = \sum_c u(c)\, H\big(u(\cdot \mid c),\, q(\cdot \mid c)\big).$$

Therefore,

$$R(q) = \sum_c \Big( p(c)\, H\big(p(\cdot \mid c), q(\cdot \mid c)\big) + \beta\, u(c)\, H\big(u(\cdot \mid c), q(\cdot \mid c)\big) \Big). \tag{7}$$

Crucially, for each fixed context $c$, the distribution $q(\cdot \mid c)$ appears *only* in the $c$-th summand. Hence, minimizing $R(q)$ over all conditionals $q(\cdot \mid c)$ reduces to minimizing each summand independently.

**Case 1: non-adjacent pairs.** If $j \neq i + 1$, then by construction of $p$ we have $p(c) = 0$. The $c$-th summand of (7) becomes $\beta\, u(c)\, H(u(\cdot \mid c), q(\cdot \mid c))$, which is minimized by $q(\cdot \mid c) = u(\cdot \mid c) = u(\cdot)$ (this also follows from Proposition A.1 with $\alpha_1 = 0$, $\alpha_2 = \beta u(c)$, and $p_2 = u(\cdot \mid c)$).

**Case 2: adjacent pairs.** If $j = i + 1$, then $p(c) = p_{\text{obs}}(i, i+1)\, p_d(Y_i = y) > 0$ and

$$p(\cdot \mid c) = p_d(\cdot \mid Y_i = y)$$

by the chain factorization (5). The $c$-th summand of (7) is

$$p(c)\, H\big(p_d(\cdot \mid Y_i = y), q(\cdot \mid c)\big) + \beta\, u(c)\, H\big(u(\cdot), q(\cdot \mid c)\big),$$

which is minimized (by Proposition A.1 with $\alpha_1 = p(c)$, $\alpha_2 = \beta u(c)$, $p_1 = p_d(\cdot \mid Y_i = y)$, and $p_2 = u(\cdot)$) by

$$q^\star(\cdot \mid c) = \lambda_{i,y}\, p_d(\cdot \mid Y_i = y) + (1 - \lambda_{i,y})\, u(\cdot), \quad \text{where} \quad \lambda_{i,y} = \frac{p(c)}{p(c) + \beta u(c)} \in (0,1).$$

Combining both cases proves the theorem. $\qquad\square$

**Theorem A.3** (Signal dilution for GALP (fluency trap)). *Fix a prompt $x$ and two candidate responses $y^{(c)}$ ("correct") and $y^{(h)}$ ("hallucinated") of equal length $m$ tokens. Let $\mathcal{I}_{logic} \subseteq \{1, \ldots, m\}$ be indices of logic-bearing tokens with $|\mathcal{I}_{logic}| = N$ and let $\mathcal{I}_{style}$ be the complement. Assume there exist constants $\delta > 0$ and $\Delta > 0$ such that, for all $t$,*

$$\log P_\theta\left(y_t^{(h)} \mid y_{<t}^{(h)}, x\right) - \log P_\theta\left(y_t^{(c)} \mid y_{<t}^{(c)}, x\right) \geq \begin{cases} \delta, & t \in \mathcal{I}_{style}, \\ -\Delta, & t \in \mathcal{I}_{logic}. \end{cases}$$

*Then the global average log-probability score (GALP) satisfies*

$$\text{GALP}(y^{(h)} \mid x) > \text{GALP}(y^{(c)} \mid x) \quad \text{whenever} \quad m > \frac{N(\Delta + \delta)}{\delta}.$$

*Proof.* By definition (Equation 1), the score difference is

$$\text{GALP}(y^{(h)} \mid x) - \text{GALP}(y^{(c)} \mid x) = \frac{1}{m} \sum_{t=1}^{m} \left( \log P_\theta(y_t^{(h)} \mid y_{<t}^{(h)}, x) - \log P_\theta(y_t^{(c)} \mid y_{<t}^{(c)}, x) \right).$$

Split the sum into logic and style indices:

$$\sum_{t=1}^{m}(\cdots) = \sum_{t \in \mathcal{I}_{style}} (\cdots) + \sum_{t \in \mathcal{I}_{logic}} (\cdots).$$

Apply the assumed bounds term-by-term. For $t \in \mathcal{I}_{style}$, each summand is at least $\delta$, so the style sum is at least $(m - N)\delta$. For $t \in \mathcal{I}_{logic}$, each summand is at least $-\Delta$, so the logic sum is at least $-N\Delta$. Therefore,

$$\text{GALP}(y^{(h)} \mid x) - \text{GALP}(y^{(c)} \mid x) \geq \frac{m - N}{m}\delta - \frac{N}{m}\Delta.$$

The right-hand side is positive iff $(m - N)\delta > N\Delta$, i.e., $m > \frac{N(\Delta+\delta)}{\delta}$. $\qquad \square$

**Theorem A.4** (Step-level robustness of LALP). *Let $y^{(c)}$ and $y^{(h)}$ be segmented into the same $p$ steps (sentences) $s_1, \ldots, s_p$. Let $\ell_i(\cdot)$ denote the (token-average) log probability of step $s_i$ under the local window used by LALP (Equation 3). Suppose there exists a "fatal" step index $i^\star$ and a gap $\gamma > 0$ such that*

$$\ell_{i^\star}(y^{(c)}) - \ell_{i^\star}(y^{(h)}) \geq \gamma \quad \text{and} \quad \ell_i(y^{(c)}) = \ell_i(y^{(h)}) \text{ for all } i \neq i^\star.$$

*Then*

$$\text{LALP}(y^{(c)} \mid x) - \text{LALP}(y^{(h)} \mid x) \geq \frac{\gamma}{p}.$$

**Dilution comparison.** Theorem A.3 shows that GALP dilutes by total tokens $m$, while Theorem A.4 shows that LALP dilutes by number of steps $p$. In our setting, $p \ll m$ (typically $\sim$100–300 steps vs. 10K+ tokens), so LALP is *far less diluted*. Note that if $p$ grows with $m$ (more sentences/steps in longer responses), the separation $\gamma/p$ does decay, but it decays with the number of *reasoning moves*, not the number of *tokens*, which is the key advantage.

*Proof.* By definition, $\text{LALP}(y \mid x) = \frac{1}{p} \sum_{i=1}^{p} \ell_i(y)$. Hence,

$$\text{LALP}(y^{(c)} \mid x) - \text{LALP}(y^{(h)} \mid x) = \frac{1}{p} \sum_{i=1}^{p} \left( \ell_i(y^{(c)}) - \ell_i(y^{(h)}) \right).$$

Under the assumptions, all terms cancel except the fatal step $i^\star$, so

$$\text{LALP}(y^{(c)} \mid x) - \text{LALP}(y^{(h)} \mid x) = \frac{1}{p}\left( \ell_{i^\star}(y^{(c)}) - \ell_{i^\star}(y^{(h)}) \right) \geq \frac{\gamma}{p}.$$

$\qquad \square$

**Corollary A.5** (Robustness with bounded non-fatal differences). *In the setting of Theorem A.4, replace the equality assumption for $i \neq i^{\star}$ with the weaker condition*

$$\ell_i(y^{(c)}) - \ell_i(y^{(h)}) \geq -\eta \quad \text{for all } i \neq i^{\star}$$

*for some $\eta \geq 0$, while keeping $\ell_{i^{\star}}(y^{(c)}) - \ell_{i^{\star}}(y^{(h)}) \geq \gamma$. Then*

$$\text{LALP}(y^{(c)} \mid x) - \text{LALP}(y^{(h)} \mid x) \geq \frac{\gamma - (p-1)\eta}{p}.$$

*In particular, if $\gamma > (p-1)\eta$, then $\text{LALP}(y^{(c)} \mid x) > \text{LALP}(y^{(h)} \mid x)$.*

*Proof.* From the identity in the proof of Theorem A.4,

$$\text{LALP}(y^{(c)} \mid x) - \text{LALP}(y^{(h)} \mid x) = \frac{1}{p} \sum_{i=1}^{p} \left( \ell_i(y^{(c)}) - \ell_i(y^{(h)}) \right).$$

Lower bound the sum by separating the fatal step $i^{\star}$ and applying the assumptions:

$$\sum_{i=1}^{p}(\cdots) = \left( \ell_{i^{\star}}(y^{(c)}) - \ell_{i^{\star}}(y^{(h)}) \right) + \sum_{i \neq i^{\star}} \left( \ell_i(y^{(c)}) - \ell_i(y^{(h)}) \right) \geq \gamma - (p-1)\eta.$$

Dividing by $p$ yields the stated bound. □

**Connection to GALP vs. LALP.** This toy model highlights why whole-trajectory/global scoring can fail in long reasoning:

- **Global scoring queries under-supported contexts**: When the model has sparse coverage at the trajectory level, its predictions collapse toward a background distribution for those configurations, making global sequence scores less informative and easier to confound by fluent scaffolding (Theorem A.2).

- **Local scoring aligns with coverage**: LALP evaluates step transitions under a short window (Equation 3), precisely where the model has dense support and nontrivial signal (the $\lambda p_d + (1-\lambda)u$ region). This is consistent with our empirical finding that teacher rankings are stable for small windows and converge to the pathological global ranking as the window grows (Figure 3).

- **Signal dilution compounds the problem**: Even when global scoring produces some signal, it is diluted by token-level averaging (Theorem A.3), while LALP's step-level aggregation preserves sensitivity to reasoning-critical moves (Theorem A.4).

**Limitations of the toy model.** We acknowledge several simplifications:

1. **Markov assumption**: The true data-generating process for reasoning may involve genuine long-range dependencies. Our model abstracts this as a chain to isolate the coverage effect; the key takeaway is about *support density*, not the structure of true dependencies.

2. **Uniform background**: In practice, the "default" when context is uninformative may not be uniform but could favor repetition/self-conditioning (the opposite of uniform). Our empirical analysis of self-reinforcement (Section 7.4) addresses this complementary failure mode, which the theory does not capture.

3. **Extension to window size $k$**: If the observation process exposes pairs within distance $k$, then the same argument implies a reasoning gap beyond distance $k$, matching our motivation for choosing a small local window for LALP.

# B. Dataset and evaluation details

## B.1. Step Segmentation Implementation

For decomposing responses into reasoning steps, we have tried using a deterministic sentence-based splitting heuristic. However, to improve correct step splitting with appropriate context, we adopt a capable, open-weight large language model, GLM-4.5-Air (Zeng et al., 2025a), which can handle large context that is necessary for our long reasoning data, to perform local step splitting with the following prompt:

```
Given the problem and solution to the problem, can you please split the
solution into groups of logical sub steps and return those groups in a
json format:

    problem: {problem}
    solution: {solution}

    E.g. {
    "sentence_groups": {
        "group1": [
        "sentences from solution"
        ],
        "group2": [
        "sentences from solution"
        ],
        ...
    }
    }

Please split the solution into logical steps and return those steps in a json format.
Remember do not modify the solution phrasing, keep the wordings original and
do not skip any step.
```

## B.2. Dataset Details

**Training Datasets.** For training, we use two primary datasets. First, we use the MATH dataset (Hendrycks et al., 2021), and following prior works (Zeng et al., 2025b; Yu et al., 2025b), we filter it to include only questions of difficulty levels 3-5, yielding 8,890 prompts (available at `https://huggingface.co/datasets/EleutherAI/hendrycks_math`). Second, to train models on reasoning data, we use the LIMO dataset (Ye et al., 2025), a carefully curated collection of 817 prompts (available at `https://huggingface.co/datasets/GAIR/LIMO`).

**Evaluation Datasets.** To evaluate the performance of the model in mathematical capabilities, we include a wide suite of math benchmarks, including:

- MATH500 (Hendrycks et al., 2021) (500 Samples)
  URL: `https://huggingface.co/datasets/EleutherAI/hendrycks_math`

- AIME 2025 (American Invitational Mathematics Examination) (30 Samples)
  URL: `https://huggingface.co/datasets/opencompass/AIME2025`

- AMC 2023(American Mathematics Competition) (40 Samples)
  URL: `https://huggingface.co/datasets/math-ai/amc23`

- MINERVA (Lewkowycz et al., 2022) (272 Samples)
  URL: `https://huggingface.co/datasets/knoveleng/Minerva-Math`

- KAOYAN (Chinese Graduate School Entrance Examinations) (199 Samples)
  URL: `https://github.com/GAIR-NLP/LIMO/blob/main/eval/data/kaoyan/test.jsonl`

- OLYMPIADBENCH (He et al., 2024) (675 Samples)
  URL: `https://huggingface.co/datasets/knoveleng/OlympiadBench`

- CN_MATH_2024 (Chinese High School Mathematics League Competition) (30 Samples)
  URL: `https://github.com/GAIR-NLP/LIMO/blob/main/eval/data/cn_math_2024/test.jsonl`

- GPQA-D (A Graduate-Level Google-Proof Q&A Benchmark) (198 Samples)
  URL: `https://huggingface.co/datasets/Idavidrein/gpqa`

- LCBv2 (LiveCodeBench) (511 Samples)
  URL: `https://github.com/LiveCodeBench/LiveCodeBench`

## B.3. Model Details

**Student Models.** For student models, we perform supervised fine-tuning on:

- Qwen2.5-Math-7B (Yang et al., 2024c)
  URL: `https://huggingface.co/Qwen/Qwen2.5-Math-7B`

- Qwen2.5-7B-Instruct (Yang et al., 2024a;b)
  URL: `https://huggingface.co/Qwen/Qwen2.5-7B-Instruct`

- Qwen2.5-32B-Instruct (Yang et al., 2024a;b)

- Llama-3.1-8B-Instruct (Grattafiori et al., 2024)
  URL: `https://huggingface.co/meta-llama/Llama-3.1-8B-Instruct`

**Teacher Models.** For teacher models, we sample responses from the following models:

- Qwen2.5-72B-Instruct (Yang et al., 2024a;b)
  URL: `https://huggingface.co/Qwen/Qwen2.5-32B-Instruct`

- Gemma3-27B-IT (Team et al., 2025)
  URL: `https://huggingface.co/google/gemma-3-27b-it`

- DeepSeek-R1 (DeepSeek-AI et al., 2025)
  URL: `https://huggingface.co/deepseek-ai/DeepSeek-R1`

- QWQ-32B(Team, 2025c)
  URL: `https://huggingface.co/Qwen/QWQ32b`

- Qwen3-32B (Yang et al., 2025a)
  URL: `https://huggingface.co/Qwen/Qwen3-32B`

## B.4. Experimental Details

**Sampling Hyperparameters.** Training data for fine-tuning student models were generated by sampling outputs from teacher models. We use the vLLM library (Kwon et al., 2023) for this process to ensure efficient inference, employing the sampling hyperparameters detailed in Table 5.

**Training Hyperparameters.** For supervised fine-tuning on student models, we leverage the LLaMA-Factory (Zheng et al., 2024) platform that offers efficient training and apply the following setting of hyperparameters (listed in Table 6):

| Property | Value |
|---|---|
| Number of samples | 1/16 |
| Temperature | 0.0/1.0 |
| Top P | 1.0/0.95 |
| Top K | 1/40 |
| Max Tokens | 42786+ |

*Table 5.* The hyperparameters for sampling from the teacher models using vLLM (Kwon et al., 2023).

| Property | Value |
|---|---|
| Train Batch Size Per Device | 1/2 |
| Gradient Accumulation Steps | 8 |
| Learning Rate | $5.0 \times 10^{-6}/1.0 \times 10^{-5}$ |
| Epochs | 10/15 |
| Warmup Ratio | 0.0 |
| BFloat16 | True |

*Table 6.* The hyperparameters for SFT the student models using LLaMA Factory (Zheng et al., 2024).

**Evaluation Hyperparameters.** After models are trained, we evaluate the models on a variety of mathemtical benchmarks using the evaluation library from LIMO (Ye et al., 2025) (URL: `https://github.com/GAIR-NLP/LIMO/tree/main/eval`) with the following hyperparameters (Table 7):

After training, we evaluate the models on a range of mathematical benchmarks using the evaluation library provided by LIMO (Ye et al., 2025) based on the Qwen2.5-Math evaluation code (Yang et al., 2024b) (available at `https://github.com/GAIR-NLP/LIMO/tree/main/eval`). The evaluation is conducted using the hyperparameter settings from DeepSeek-R1 (DeepSeek-AI et al., 2025) as detailed in Table 7.

| Property | Value |
|---|---|
| Temperature | 0.0/0.6 |
| Max Tokens | 32768 |
| Top P | 1/0.95 |
| Pass@K | 1/8 |
| Samples | 1/8 |

*Table 7.* The hyperparameters for evaluation of the student models at the inference stage using evaluation code from Qwen2.5-Math evaluation (Yang et al., 2024b).

**Software and Hardware.** In our experiments we used NVIDIA 4xA100 GPUs for training and evaluation. For reproducibility of our results, we share our code in an anonymized repository for the submission purposes: https://anonymous.4open.science/r/lalp-5272/.

## C. Additional Results

### C.1. Loss Comparison

We compare the loss curves of student models trained on data selected using global and local log likelihood criteria, as summarized in Table 3. The corresponding loss plots are presented in Figure 6.

As shown in Figure 6, models trained on responses with the highest global log-likelihood demonstrate the fastest convergence

and lowest training loss compared to those trained on randomly selected data or responses with the highest local log-likelihood. This behavior is expected, as high global log-likelihood responses likely represent more cohesive and natural samples as a whole, which align more closely with the student model's existing representation space. Such data may provide clearer learning signals, enabling the model to fit the training distribution more efficiently. However, as shown earlier in Table 3, instead, the model trained on data selected by highest local log-likelihood ultimately achieves better downstream performance. This highlights a key insight: while global log-likelihood data may facilitate faster convergence during training, this does not necessarily translate to better generalization, underscoring the limitations of relying solely on loss curves as indicators of final model performance.

## C.2. Data Composition From Selection

In Section 7, we have chosen LIMO responses across three teachers (DeepSeek-R1, QWQ-32B, Qwen3-32B) based on local and global naturalness and provided results in Table 3. Here, we provide the composition of selected responses across teachers depending on the method in Table 8.

| | DeepSeek-R1 | QWQ-32B | Qwen3.0-32B |
|---|---|---|---|
| **Student: Qwen2.5-32B-Instruct** | | | |
| Random | 33.3 | 33.3 | 33.4 |
| Local Lowest | 42.4 | 11.3 | 46.3 |
| Global Highest | 47.6 | 7.2 | 45.2 |
| Local Highest | 42.4 | 36.3 | 21.3 |
| **Student: Qwen2.5-32B-Instruct** | | | |
| Random | 33.3 | 33.3 | 33.4 |
| Local Lowest | 43.3 | 20.4 | 36.3 |
| Global Highest | 47.2 | 8.6 | 44.2 |
| Local Highest | 26.8 | 44.9 | 28.3 |

*Table 8.* Data composition from different teacher models for the LIMO responses depending on the selection method(%).

## C.3. Additional Examples on Self-Conditioning

**Example 3: Multi-digit number recall ("1228").** A more dramatic example: the model computes $(2457 - 1)/2 = 1228$ and subsequently references this value:

| Context | Log prob of "1" | Probability |
|---|---|---|
| First calculation: "$= 1228$" | $-0.33$ | 71% |
| Recall: "is 1228 divisors" | $-0.00006$ | 99.99% |
| Usage: "1228 - 639 = ..." | $-0.006$ | 99.4% |

The model transitions from *calculating* (71% confidence) to *copying* (99.99% confidence). Under global scoring, all three occurrences contribute to the average log probability, but only the first reflects genuine reasoning difficulty.

**Example 4: Stylistic self-conditioning ("come in pairs").** Self-conditioning affects not only numbers but also phrasing choices:

| Context | Log prob of "come" | Probability |
|---|---|---|
| First use: "divisors ... come in pairs" | $-1.28$ | 27% |
| Second use: "divisors come in pairs" | $-0.66$ | 51% |

The model nearly doubles its confidence in the word "come" simply because it used that phrasing earlier. This stylistic self-conditioning inflates GALP scores for responses with repetitive phrasing patterns.

| | MATH | AIME25 | AMC | MINERVA | KAOYAN | OLYMPIADB | CN_MATH24 | AVG | GALP | LALP |
|---|---|---|---|---|---|---|---|---|---|---|
| **Student: Qwen2.5-7B-Instruct** | | | | | | | | | | |
| **Student Before SFT** | 0.752 | 0.167 | 0.500 | 0.268 | 0.216 | 0.404 | 0.167 | 0.353 | - | - |
| **Qwen3-32B Data** | 0.714 | 0.166 | 0.500 | 0.279 | 0.389 | 0.375 | 0.133 | 0.365 | -0.697 | -0.279 |
| **DeepSeek-R1 Data** | 0.784 | 0.166 | 0.600 | 0.239 | 0.330 | 0.441 | 0.233 | 0.399 | -0.796 | -0.264 |
| **QWQ-32BData** | 0.780 | 0.266 | 0.600 | 0.275 | 0.356 | 0.442 | 0.2 | 0.417 | -0.743 | -0.241 |
| **Student: Qwen2.5-32B-Instruct** | | | | | | | | | | |
| **Student Before SFT** | 0.822 | 0.133 | 0.700 | 0.298 | 0.422 | 0.471 | 0.233 | 0.445 | - | - |
| **Qwen3-32B Data** | 0.882 | 0.567 | 0.900 | 0.353 | 0.598 | 0.559 | 0.600 | 0.637 | -0.800 | -0.257 |
| **DeepSeek-R1 Data** | 0.896 | 0.467 | 0.925 | 0.338 | 0.613 | 0.644 | 0.733 | 0.659 | -0.895 | -0.241 |
| **QWQ-32BData** | 0.916 | 0.633 | 0.975 | 0.364 | 0.653 | 0.689 | 0.800 | 0.719 | -0.888 | -0.218 |

*Table 9.* Teacher selection results. LALP correctly ranks teachers by downstream performance (QwQ > DeepSeek-R1 > Qwen3), while GALP produces an inverted ranking. Colors: best , middle , worst .

| | MATH | AIME25 | AMC | MINERVA | KAOYAN | OLYMPIAD | CN_MATH24 | AVG |
|---|---|---|---|---|---|---|---|---|
| **Student: Qwen2.5-7B-Instruct** | | | | | | | | |
| **Original Model** | 0.752 | 0.167 | 0.500 | 0.268 | 0.216 | 0.404 | 0.167 | 0.353 |
| **Random** | 0.768 | 0.133 | 0.625 | 0.268 | 0.367 | 0.456 | 0.233 | 0.407 |
| **GALP** | 0.762 | 0.2 | 0.6 | 0.268 | 0.381 | 0.441 | 0.233 | 0.412 |
| **Local Lowest** | 0.742 | 0.167 | 0.575 | 0.298 | 0.342 | 0.433 | 0.233 | 0.399 |
| **LALP** | 0.788 | 0.2 | 0.625 | 0.298 | 0.392 | 0.441 | 0.333 | **0.440** |
| **Student: Qwen2.5-32B-Instruct** | | | | | | | | |
| **Original Model** | 0.824 | 0.133 | 0.700 | 0.298 | 0.422 | 0.471 | 0.233 | 0.445 |
| **Random** | 0.906 | 0.400 | 0.925 | 0.327 | 0.628 | 0.636 | 0.733 | 0.651 |
| **GALP** | 0.876 | 0.433 | 0.825 | 0.331 | 0.592 | 0.636 | 0.733 | 0.632 |
| **Local Lowest** | 0.896 | 0.400 | 0.825 | 0.324 | 0.608 | 0.640 | 0.700 | 0.623 |
| **LALP** | 0.902 | 0.667 | 1.000 | 0.353 | 0.653 | 0.673 | 0.833 | **0.726** |

*Table 10.* Response selection from mixed-teacher pool. LALP (Local Highest) outperforms GALP (Global Highest) by **+9.4%** on the 32B student (0.726 vs 0.632). Notably, LALP-selected data even outperforms training on all responses from the best single teacher (QwQ-32B: 0.719 in Table 2).

## C.4. Breakdown of results in main paper.

## C.5. Additional Results on MATH prompts

We present additional results for Qwen2.5-7B-Instruct on the MATH benchmark, using responses generated by two different teacher models: Qwen2.5-72B-Instruct, which tends to produce shorter responses, and QWQ-32B, which generates longer reasoning responses. We provide a comprehensive summary of these results in Table 14.

## C.6. Ablation: Test Set Coverage

Here we present other test sets that also show the gap between GALP and LALP behavior in Figures 7 and 8.

## C.7. Ablation: Context Window Size vs. Performance

LALP uses a local context window when scoring each reasoning step. This window controls the trade-off between providing enough preceding context to make a step interpretable and avoiding the self-conditioning effects of full-trajectory scoring. In the main analysis, we show that small local windows produce teacher rankings that better match downstream performance, whereas larger windows increasingly recover the GALP failure mode. Here, we further study whether the context window size also affects downstream fine-tuning performance.

Figure 9 reports the ablation in the math-domain. We observe that performance is best when the window contains a moderate amount of local context. Very small windows can under-condition the step, making some transitions difficult to evaluate. This supports the view that LALP benefits from preserving locality rather than simply truncating the context as aggressively as possible.

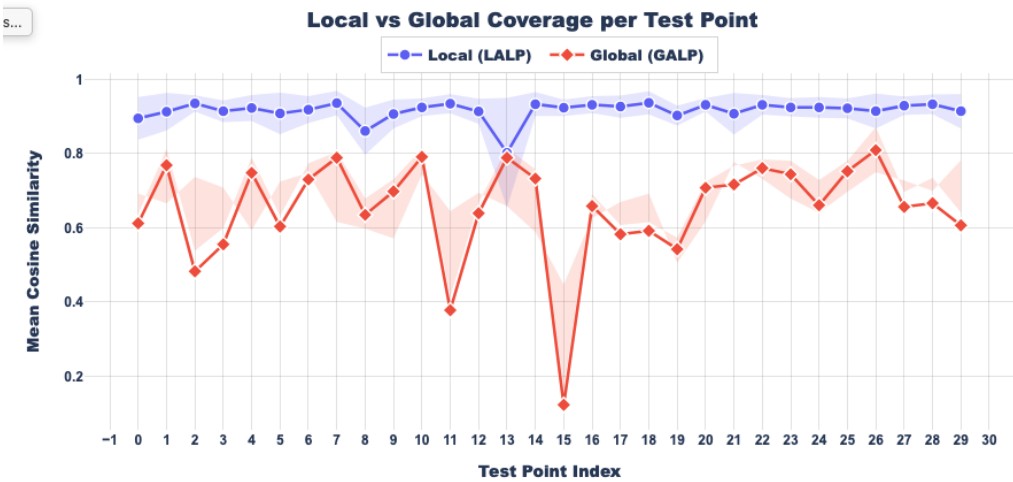

*Figure 7.* AMC Test Points

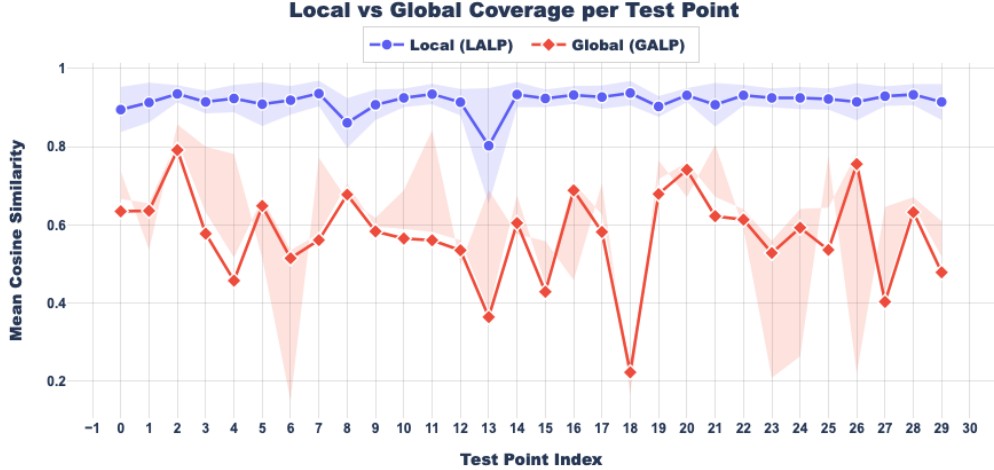

*Figure 8.* GSM8K Test Points

To test whether this behavior is specific to mathematics, we also run a context-window ablation in the coding domain. We generate responses for 5000 prompts from OpenCodeReasoning and LeetCode, use the selected data to fine-tune the Qwen student models, and evaluate on LiveCodeBench. The results are shown in Table 11.

Table 11 shows that the benefit of LALP is not specific to the math-only setting. Across both student sizes, LALP outperforms GALP and Random over a range of local context sizes. The smallest window, $k = 1$, is weaker, suggesting that a preceding step may not provide enough context to reliably evaluate the next reasoning transition reliably. Moderate local windows perform best overall, and the best value varies slightly between student sizes.

These results suggest that LALP is not highly sensitive to a single hand-picked context window. Rather, its gains appear across a range of local windows that provide enough immediate context while avoiding full-trajectory conditioning. This cross-domain ablation is consistent with the step-compositional view of reasoning: useful supervision is better captured by locally supported reasoning transitions than by the likelihood of an entire response under a long accumulated prefix.

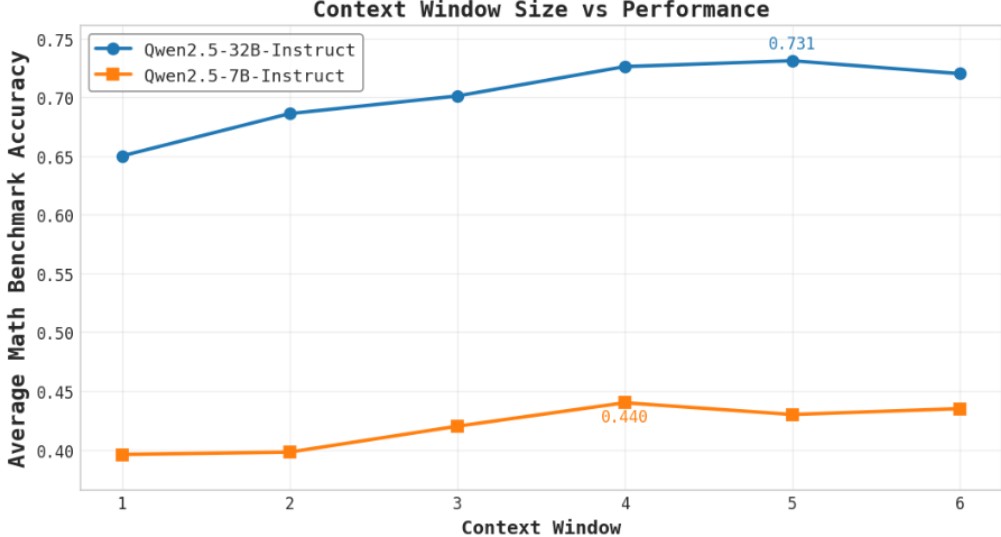

*Figure 9.* Context window size vs performance. Ablation on two student models.

*Table 11.* Context-window ablation in the coding domain. All LALP variants use the same scoring rule and differ only in the number of preceding steps used as local context.

| Selection Method | 32B LCB Avg | 7B LCB Avg |
|---|---|---|
| LALP ($k = 1$) | 0.561 | 0.334 |
| LALP ($k = 2$) | 0.579 | 0.342 |
| LALP ($k = 3$) | 0.578 | 0.345 |
| LALP ($k = 4$) | **0.589** | 0.350 |
| LALP ($k = 5$) | 0.576 | **0.352** |
| GALP | 0.555 | 0.310 |
| Random | 0.527 | 0.291 |

### C.8. Qualitative Examples: Global vs Local Nearest Neighbors

To provide intuition for the quantitative coverage results, we present qualitative nearest-neighbor examples comparing trajectory-level (global) and step-level (local) embeddings from AIME 2025 test problems.

**Global (Trajectory-Level) Examples.** When matching *full solution trajectories*, the nearest training neighbors often differ substantially in problem structure and reasoning approach. This reflects the sparse manifold coverage (cosine similarity 0.760) reported in Figure 2.

| AIME Test Trajectory (truncated) | Nearest Training Trajectory (truncated) |
|---|---|
| "Let me work through this combinatorics problem systematically. We need to count the number of ways to arrange..." (cos=0.72) | "I'll solve this number theory problem step by step. First, let's identify the prime factorization..." |
| "This is a geometry problem involving circles and tangent lines. Let me set up coordinates..." (cos=0.68) | "For this probability question, I need to compute the expected value by considering all possible outcomes..." |

*Table 12.* Global nearest-neighbor examples: full trajectories match poorly across problem types.

**Local (Step-Level) Examples.** In contrast, when matching *individual reasoning steps*, nearly every test step finds a highly similar training step. This reflects the dense manifold coverage (cosine similarity 0.935) that enables compositional generalization.

| AIME Test Step | Nearest Training Step |
|---|---|
| "Since $\gcd(a, b) = 1$, we can apply Bézout's identity to find integers $x, y$ such that $ax + by = 1$." (cos=0.96) | "Because $\gcd(m, n) = 1$, Bézout's lemma guarantees integers $u, v$ with $mu + nv = 1$." |
| "Substituting $x = 2$ into the equation gives $f(2) = 4 + 3(2) - 7 = 3$." (cos=0.94) | "Plugging in $t = 2$ yields $g(2) = 2^2 + 5(2) - 4 = 10$." |
| "By the triangle inequality, $|a + b| \leq |a| + |b|$." (cos=0.98) | "The triangle inequality gives us $|x + y| \leq |x| + |y|$." |
| "Let $S_n$ denote the sum of the first $n$ terms." (cos=0.97) | "Define $T_k$ as the sum of the first $k$ elements." |

*Table 13.* Local nearest-neighbor examples: individual reasoning steps match closely across problems, enabling compositional reuse.

**Interpretation.** These examples illustrate why step-level data selection outperforms trajectory-level selection. While complete solutions to novel AIME problems are genuinely new (no close training match exists), the *atomic reasoning moves*, such as algebraic substitutions, theorem applications, definitional setups, are well-represented in training data. LALP leverages this compositional structure by scoring responses based on how well-supported each individual step is, rather than requiring a global template match that rarely exists.

### C.9. GALP Works Within Single Teacher (Short Responses)

We replicate the finding from prior work (Zhang et al., 2025) that global log probability selection works well within a single teacher on shorter responses. This establishes the baseline validity of GALP before examining its failure modes.

**Setup.** We use MATH prompts of level 3-5 difficulty (8,890 prompts). For each prompt, we generate 16 responses using a teacher model with temperature 0.6 and top-p 0.95. We select responses with highest, middle, and lowest global log probability for comparison. Students are fine-tuned for 5 epochs with learning rate 1e-5.

| | | MATH | AIME25 | AMC | MINERVA | KAOYAN | OLYMPIADB | CN_MATH24 | AVG |
|---|---|---|---|---|---|---|---|---|---|
| | | | | | **Student: Qwen2.5-7B-Instruct** | | | | |
| | Original Model | 0.752 | 0.167 | 0.5 | 0.268 | 0.216 | 0.404 | 0.167 | 0.353 |
| **Teacher:** | Lowest LP | 0.678 | 0.1 | 0.3 | 0.224 | 0.296 | 0.314 | 0.133 | 0.292 |
| **Qwen2.5-72B** | Middle LP | 0.71 | 0.1 | 0.425 | 0.257 | 0.336 | 0.339 | 0.2 | 0.338 |
| **-Instruct** | Highest LP | 0.744 | 0.133 | 0.5 | 0.252 | 0.391 | 0.391 | 0.167 | 0.368 |
| **Teacher:** | Lowest LP | 0.667 | 0.1 | 0.375 | 0.165 | 0.226 | 0.29 | 0.133 | 0.279 |
| **Gemma3-27B** | Middle LP | 0.716 | 0.1 | 0.475 | 0.129 | 0.246 | 0.357 | 0.167 | 0.313 |
| **-IT** | Highest LP | 0.712 | 0.1 | 0.5 | 0.176 | 0.251 | 0.362 | 0.133 | 0.319 |
| | | | | | **Student: Qwen2.5-Math-7B** | | | | |
| | Original Model | 0.5 | 0.033 | 0.425 | 0.092 | 0.1 | 0.164 | 0.133 | 0.207 |
| **Teacher:** | Lowest LP | 0.77 | 0.033 | 0.5 | 0.26 | 0.407 | 0.381 | 0.1 | 0.350 |
| **Qwen2.5-72B** | Middle LP | 0.79 | 0.1 | 0.55 | 0.25 | 0.41 | 0.416 | 0.133 | 0.378 |
| **-Instruct** | Highest LP | 0.778 | 0.133 | 0.6 | 0.35 | 0.46 | 0.398 | 0.167 | 0.412 |
| **Teacher:** | Lowest LP | 0.802 | 0.133 | 0.525 | 0.213 | 0.331 | 0.436 | 0.2 | 0.377 |
| **Gemma3-27B** | Middle LP | 0.792 | 0.1 | 0.575 | 0.246 | 0.312 | 0.45 | 0.367 | 0.406 |
| **-IT** | Highest LP | 0.816 | 0.167 | 0.625 | 0.25 | 0.387 | 0.455 | 0.433 | 0.448 |

*Table 14.* GALP selection within single teacher (short responses). Highest LP consistently outperforms lowest LP within each teacher, confirming that GALP works in this controlled setting. However, rankings do not transfer across teachers (e.g., lowest LP from Qwen2.5-72B gives 0.292 while lowest LP from Gemma3-27B gives 0.279 for different underlying reasons).

**Observations.** Table 14 shows that within each teacher, highest-GALP selection consistently outperforms lowest-GALP selection. Figure 10 confirms a positive within-teacher correlation. However, the cross-teacher rankings do not align: responses with similar global LP from different teachers yield different performance. This foreshadows the failure mode we examine in the main text, where mixing diverse teachers causes GALP to break down entirely.

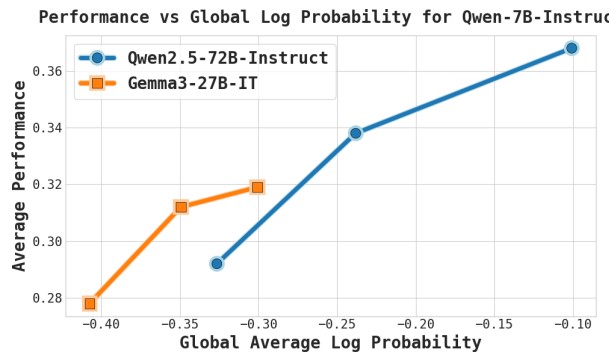 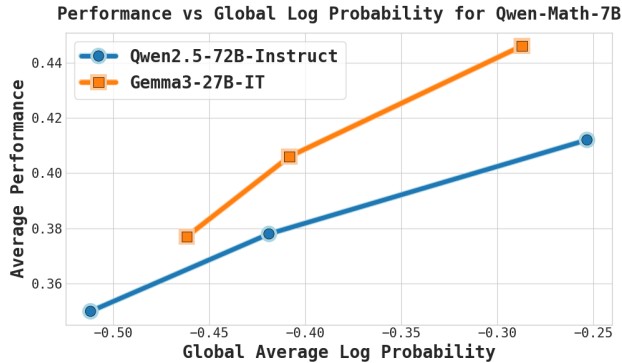

*Figure 10.* Within-teacher correlation between GALP and performance. Within each teacher (same color), higher global LP correlates with better performance. However, the lines do not align across teachers, indicating GALP cannot reliably compare responses from different sources.

|  |  | MATH | AIME25 | AMC | MINERVA | KAOYAN | OLYMPIADB | CN_MATH24 | AVG |
|---|---|---|---|---|---|---|---|---|---|
| | | **Student: Qwen2.5-7B-Instruct** | | | | | | | |
| | Original Model | 0.752 | 0.167 | 0.500 | 0.268 | 0.216 | 0.404 | 0.167 | 0.353 |
| **Teacher:** | Lowest GALP | 0.678 | 0.100 | 0.300 | 0.224 | 0.296 | 0.314 | 0.133 | 0.292 |
| **Qwen2.5-72B** | Middle GALP | 0.710 | 0.100 | 0.425 | 0.257 | 0.336 | 0.339 | 0.200 | 0.338 |
| **-Instruct** | Highest GALP | 0.744 | 0.133 | 0.500 | 0.252 | 0.391 | 0.391 | 0.167 | 0.368 |
| **Teacher:** | Lowest GALP | 0.686 | 0.100 | 0.400 | 0.246 | 0.286 | 0.324 | 0.200 | 0.320 |
| **QwQ-32B** | Middle GALP | 0.710 | 0.167 | 0.425 | 0.272 | 0.336 | 0.333 | 0.200 | 0.349 |
| | Highest GALP | 0.732 | 0.167 | 0.475 | 0.279 | 0.412 | 0.382 | 0.233 | 0.382 |

*Table 15.* Within-teacher sanity check for GALP. For each teacher, responses are partitioned into terciles by the pre-SFT student's global average log probability (Eq. 1). Fine-tuning on the highest-GALP tercile yields the strongest average downstream performance.

## C.10. Extended Results for Within-teacher Sanity Check for GALP

We provide a benchmark breakdown of results for the GALP selection within a single teacher in Table 15.

## C.11. Generalizability Experiments

To explicitly test this generalizability, we have since conducted additional experiments in general science. For the models trained on math data from our paper, we computed performance the GPQA-Diamond benchmark, which tests expert-level reasoning across biology, physics, and chemistry. We obtained the following results:

*Table 16.* GPQA-Diamond Benchmark Results

| Model / Method | GPQA-Diamond (pass@1) |
|---|---|
| Original Qwen2.5-32B-Instruct | 0.551 |
| All 3 Teachers' Responses | 0.439 |
| LIMO-32B | 0.626 |
| Sky-T1-32B-Preview | 0.566 |
| OpenThinker2-32B | 0.646 |
| Global Highest (GRAPE) | 0.611 |
| Local Highest (Ours) | 0.702 |

Notably, our LALP selection method not only surpasses the Global Highest baseline but also outperforms other state-of-the-art models. This is particularly significant as these other models were trained on substantially larger and more diverse long reasoning datasets, underscoring the efficiency and effectiveness of our data curation technique.

To further demonstrate the generalizability of our approach, we extended our evaluation to code reasoning. We generated responses for 5,000 prompts from the OpenCodeReasoning and LeetCode datasets and used them to fine-tune the Qwen2.5-

32B-Instruct model. The models performance were then evaluated on the LiveCodeBench v2 benchmark.

| Method | LiveCodeBench-easy | LiveCodeBench-medium | LiveCodeBench-hard |
|---|---|---|---|
| Original Qwen2.5-32B-Instruct | 0.890 | 0.471 | 0.114 |
| Global Highest (GRAPE) | 0.845 | 0.588 | 0.232 |
| Local Highest (Ours) | 0.874 | 0.633 | 0.261 |

*Table 17.* LiveCodeBench v2 Benchmark Results

As the results indicate, the student model trained on data selected via LALP consistently outperforms the one trained using the global log-probability baseline across the medium and hard difficulty tiers.

These additional results from the scientific and coding domains strongly suggest that the core principle of LALP, step-level standalone justifiability, is not confined to mathematics. The method's effectiveness in identifying high-quality reasoning data appears to generalize to other domains that require complex, step-by-step inference. We will include these findings in the paper to provide a more comprehensive evaluation of our method's applicability.

### C.12. Ablation: Llama-3.1-8B-Instruct

We provide results on another student model to show the generalizability of our method in Table 18.

| | MATH | AIME25 | AMC | MINERVA | KAOYAN | OLYMPIAD | CN_MATH24 | GPQA |
|---|---|---|---|---|---|---|---|---|
| **Student: Llama-3.1-8B-Instruct** | | | | | | | | |
| **Original Model** | 0.726 | 0.0 | 0.45 | 0.316 | 0.216 | 0.361 | 0.3 | 0.656 |
| **All 3 Teachers** | 0.794 | 0.167 | 0.725 | 0.335 | 0.432 | 0.51 | 0.2 | 0.878 |
| **Global Highest** | 0.796 | 0.133 | 0.625 | 0.371 | 0.437 | 0.48 | 0.2 | 0.833 |
| **Local Highest (Ours)** | 0.814 | 0.167 | 0.8 | 0.368 | 0.467 | 0.49 | 0.233 | 0.883 |

*Table 18.* Performance of Llama-3.1-8B-Instruct student models on LIMO prompts when fine-tuned with responses from different selection strategies. The log probabilities (LP) are the global and local average log probabilities of the student model with non-greedy decoding with temperature 0.6, top-p 0.95 and over 8 samples, pass@8.

### C.13. Cross Teacher Selection: Performance with Non-Greedy Decoding

We provide results on cross teacher selection with non-greedy decoding in Table 19.

| | MATH | AIME25 | AMC | MINERVA | KAOYAN | OLYMPIAD | CN_MATH24 | AVG |
|---|---|---|---|---|---|---|---|---|
| **Student: Qwen2.5-32B-Instruct** | | | | | | | | |
| **Original Model** | 0.826 | 0.121 | 0.715 | 0.295 | 0.416 | 0.461 | 0.237 | 0.439 |
| **All 3 Teachers** | 0.835 | 0.400 | 0.887 | 0.335 | 0.578 | 0.559 | 0.583 | 0.597 |
| **Global Highest** | 0.862 | 0.442 | 0.903 | 0.338 | 0.629 | 0.634 | 0.721 | 0.647 |
| **Local Highest (Ours)** | 0.911 | 0.662 | 0.969 | 0.361 | 0.658 | 0.675 | 0.850 | **0.727** |

*Table 19.* Performance of Qwen2.5-7B-Instruct and Qwen2.5-32B-Instruct student models on LIMO prompts when fine-tuned with responses from different selection strategies. The log probabilities (LP) are the global and local average log probabilities of the student model with non-greedy decoding with temperature 0.6, top-p 0.95 and over 8 samples.

### C.14. Performance Comparison with Other SOTA Qwen-32B Models

We compare the performance of our model against several strong open-source implementations that also fine-tune the Qwen-32B-Instruct student model on comparable or larger datasets. LIMO-32B-V1(Ye et al., 2025) (available at `https://huggingface.co/GAIR/LIMO`) is trained on the LIMO prompt set using responses exclusively from the DeepSeek-R1 teacher. Sky-T1-32B-Preview(Team, 2025a) (available at `https://huggingface.co/NovaSky-AI/Sky-T1-32B-Preview`) is trained on a 17K example dataset (`https://huggingface.co/datasets/NovaSky-AI/Sky-T1_data_17k`) generated using the QWQ-32B model. OpenThinker2-32B(Team, 2025b) (available at `https://huggingface.co/open-thoughts/OpenThinker2-32B`) is trained on a substantially larger

dataset of 1.04M samples(https://huggingface.co/datasets/open-thoughts/OpenThoughts2-1M), also generated using DeepSeek-R1 as the teacher. A detailed comparison of the results is provided in Table 20.

| | MATH | AIME25 | AMC | MINERVA | KAOYAN | OLYMPIAD | CN_MATH24 | AVG | GPQA |
|---|---|---|---|---|---|---|---|---|---|
| **Student: Qwen2.5-32B-Instruct** | | | | | | | | | |
| **Original Model** | 0.824 | 0.133 | 0.700 | 0.298 | 0.422 | 0.471 | 0.233 | 0.445 | 0.551 |
| **Global Highest** | 0.876 | 0.433 | 0.825 | 0.331 | 0.592 | 0.636 | 0.733 | 0.632 | 0.611 |
| **LIMO-32B** | 0.896 | 0.433 | 0.925 | 0.346 | 0.618 | 0.630 | 0.800 | 0.664 | 0.626 |
| **Sky-T1-32B-Preview** | 0.876 | 0.200 | 0.750 | 0.301 | 0.558 | 0.507 | 0.533 | 0.532 | 0.566 |
| **OpenThinker2-32B** | 0.922 | 0.567 | 0.900 | 0.324 | 0.648 | 0.640 | 0.833 | 0.691 | 0.646 |
| **Local Highest (Ours)** | 0.902 | 0.667 | 1.000 | 0.353 | 0.653 | 0.673 | 0.833 | **0.726** | 0.694 |

*Table 20.* A performance comparison of our model with other open-source SOTA models fine tuned on the Qwen2.5-32B-Instruct student model.

## C.15. Ablation: Splitting Steps and Other Baselines

LALP requires partitioning each candidate response into reasoning steps before applying local scoring. In the main experiments, we use GLM-4.5-Air for this segmentation because it provides semantically meaningful step boundaries for long-form reasoning traces. However, relying on an external LLM segmenter may introduce additional dependencies and possible segmenter-specific bias. We therefore ablate the segmentation procedure using simpler rule-based alternatives.

We compare three segmentation strategies. The first is the GLM-4.5-Air segmenter used in the main experiments. The second uses Python's NLTK sentence tokenizer, with additional post-processing to avoid splitting inside common LaTeX expressions and displayed equations. The third uses double-newline boundaries, denoted as \n\n, which approximates paragraph-level splitting. All three variants use the same LALP scoring rule and differ only in how candidate responses are divided into local units.

We also include two heuristic baselines that select responses based only on the number of NLTK-derived steps: MOST STEPS and LEAST STEPS. These baselines test whether LALP's gains can be explained simply by a preference for more verbose or more concise traces, rather than by the local learnability signal itself.

Table 21 shows that LALP is robust to the choice of segmenter. Although GLM-4.5-Air yields the best performance, the simpler NLTK-based variant performs close to it and still clearly outperforms GALP and Random selection. The double-newline variant is slightly weaker, likely because different teachers use paragraph spacing with different frequencies, making paragraph boundaries a less stable proxy for reasoning steps across heterogeneous sources.

The verbosity baselines further show that LALP is not merely selecting longer or shorter responses. Selecting the response with the most steps improves over GALP for the 32B student but underperforms LALP and performs poorly for the 7B student. Selecting the response with the fewest steps does not provide a consistent advantage. These results suggest that LALP's gains come primarily from local step-level scoring rather than from a simple preference for response length, number of steps, or a specific LLM-based segmenter.

## C.16. Ablation: Reasoning and Non-Reasoning Teacher Pool

The main mixed-teacher experiments use three reasoning teacher models that are capable of producing long-form reasoning traces: DeepSeek-R1, Qwen3-32B-Instruct, and QwQ-32B. In this setting, we expand the response pool from three teachers to five teachers by adding Qwen2.5-72B-Instruct and Gemma3-27B-IT. These additional models increase the diversity of candidate responses and introduce traces that are not necessarily optimized for long-form mathematical reasoning. We keep the same response-selection protocol as in the main experiments.

Table 22 shows that expanding the teacher pool makes selection more challenging for the non-local baselines. Random selection drops because the enlarged pool contains a wider range of response styles and reasoning quality. GALP also remains substantially below LALP, suggesting that global response likelihood is still vulnerable to the same failure mode observed in the three-teacher setting. It can prefer responses that are fluent or natural under the student model without necessarily providing the most useful reasoning supervision. In contrast, LALP maintains strong performance for both student sizes.

*Table 21.* Ablation over step segmentation strategies and verbosity-based baselines for mixed-teacher response selection. All LALP variants use the same local scoring rule and differ only in the segmentation method.

| Selection Method | 32B Avg Math | 7B Avg Math |
| --- | --- | --- |
| LALP (GLM-4.5-Air, ours) | **0.726** | **0.440** |
| LALP (NLTK) | 0.719 | 0.431 |
| LALP ($\backslash$n$\backslash$n) | 0.712 | 0.424 |
| GALP | 0.632 | 0.412 |
| Random | 0.651 | 0.407 |
| Most Steps | 0.686 | 0.392 |
| Least Steps | 0.642 | 0.408 |

*Table 22.* Ablation with an expanded five-teacher response pool containing both reasoning-oriented and general instruction-tuned models. LALP remains robust as the candidate pool becomes more heterogeneous.

| Selection Method | 32B Avg Math | 7B Avg Math |
| --- | --- | --- |
| LALP (ours) | **0.726** | **0.440** |
| GALP | 0.641 | 0.405 |
| Random | 0.594 | 0.389 |

