# OpenReview forum: "The Signal is in the Steps: Local Scoring for Reasoning Data Selection"
_ICML.cc/2026/Conference — ICML 2026 spotlight_

### Official Review · Reviewer_Z7Ai · 2026-03-13

**Soundness:** 3
**Presentation:** 2
**Significance:** 3
**Originality:** 2
**Overall Recommendation:** 5
**Confidence:** 4

**Summary:**

This paper investigates the long-form data selection problem in reasoning tasks for LLMs. The authors point out that the prevalent GALP scoring fails due to the side effects of long-range contexts (such as "fluency trap" and "self-conditioning"), thereby overshadowing local reasoning signals. To address this issue, the paper proposes the LALP method. By segmenting responses into logical steps, LALP evaluates the logical validity of each step using only a localized context window. Experimental results demonstrate that LALP enables more precise teacher model selection prior to fine-tuning, significantly enhances the performance accuracy of downstream models and exhibits strong cross-domain generalization capabilities.

**Compliance With Llm Reviewing Policy:**

Affirmed.

**Final Justification:**

This paper addresses the "fluency trap" in long-form reasoning data selection for LLMs, where global log-probability (GALP) scores often prioritize linguistic smoothness over logical correctness. To solve this, the authors propose LALP (Local Average Log Probability), which evaluates the validity of individual logical steps within a localized context window.

The submission is recommended for Acceptance based on its strong empirical results across benchmarks like AIME and GPQA. The work is both original and highly significant, offering an efficient, lightweight strategy for distilling reasoning capabilities, a crucial task for modern model training. During the rebuttal, the authors successfully addressed concerns regarding segmentation bias by demonstrating that LALP works effectively even with simple heuristic rules (e.g., NLTK). They also proved the generalizability of their context window settings across diverse domains and polished the paper’s presentation. While the current focus is on SFT, the potential of LALP as a zero-shot Process Reward Model (PRM) for RL provides a promising future direction. Overall, this is a technically sound and practical contribution to the field of NLP.

**Key Questions For Authors:**

1. LALP currently relies heavily on a specific model (GLM-4.5-Air) to segment logical steps. Considering the inherent distribution biases in the output styles of different models, how sensitive is the proposed method to different segmentation strategies? For example, how does the performance change when using other models or simple heuristic rules (e.g., using "\n\n")?
2. Does the optimal context window size remain consistent across datasets with different distributions? Could the authors provide ablation studies on the window size across diverse domains to demonstrate LALP's generalizability?
3. Can the local step-level scoring mechanism of LALP be integrated into the RL post-training phase? Have the authors conducted any preliminary experiments or theoretical explorations in this direction?

**Limitations:**

yes

**Strengths And Weaknesses:**

# Strengths
1. The paper conducts empirical validations across multiple mainstream reasoning and scientific benchmarks, including AIME, MATH, and GPQA. The experiments encompass student models of varying parameter scales and teacher models with diverse generation styles, providing highly robust data support. Furthermore, the authors perform a profound token-level attribution analysis that elegantly quantifies the shift in probability mass, and honestly discuss the limitation that their method cannot verify long-range consistency at the end of the paper.
2. When elucidating complex mechanisms, the paper successfully abstracts core concepts such as the "fluency trap" and "self-conditioning". By utilizing intuitive qualitative examples alongside clear comparisons of probability changes, the methodology becomes highly tangible. The overall narrative is well-structured and easy for readers to follow.
3. Given the widespread current demand for distilling long-form reasoning data from frontier models, this paper offers an efficient and lightweight data selection strategy. LALP demonstrably enhances the performance of downstream fine-tuning tasks, providing valuable practical guidance for optimizing data curation pipelines in both academia and industry.
4. The authors successfully break the conventional inertia of relying on "global sequence logprobs" for data quality assessment, introducing the novel criterion of "step-level sequence logprobs". Coupled with theoretical model derivations, this work delivers a valuable new perspective for evaluating the quality of long-form reasoning data.

# Weaknesses
1. LALP relies heavily on an external large language model (GLM-4.5-Air) to segment logical steps. This inevitably introduces the splitting model's inherent biases, yet the paper lacks ablation studies regarding the robustness of different splitting strategies or granularities.
2. The ablation study assessing the impact of the context window size on performance is currently exclusively conducted on the LIMO dataset. The absence of generalization validation in other domains considerably weakens the universality of the conclusions drawn for this core hyperparameter.
3. The current mainstream paradigm for enhancing complex reasoning capabilities in LLMs is the two-stage post-training pipeline of "SFT + RL". This paper solely validates the data selection effectiveness of LALP in the SFT stage, while lacking exploration into its applicability during the RL stage. This omission limits the overarching impact of the method within the most cutting-edge reasoning model training paradigms.
4. Almost none of the figures in the submission are vector graphics, resulting in noticeable blurriness, and the font sizes in several key figures and tables (e.g., Figure 2, Table 2, Table 3) are too small; Equations require further adjustment (e.g., the formulation of LocalLP and Equation 4 ); There are still incomplete and ungrammatical sentences (e.g., "...aggregate at the We formalize this as Local Average Log Probability...")

---

> ### Author Rebuttal · Authors · 2026-03-31
>
> Thank you for your rigorous review. We deeply appreciate your push to validate our hyperparameters across domains and to contextualize our method within the broader SFT + RL paradigm. We have addressed your concerns below and in our revised manuscript.
>
> ---
> ## **W1 & Q1: Sensitivity to segmentation strategies and model bias**
>
> We agree that minimizing external dependencies and mitigating the inherent biases of the splitting model is crucial. To test LALP's robustness, we conducted additional experiments using simple, rule-based segmentation strategies:
> 1. **NLTK:** Python's NLTK sentence tokenizer (with custom post-processing for LaTeX blocks).
> 2. **Newline:** Double newline (`\n\n`) splitting.
>
> LALP rankings based on these heuristic splits showed strong Spearman correlations with our primary GLM-4.5-Air rankings:
> * **Qwen2.5-32B-Instruct:** $\rho > 0.92$ (NLTK) and $\rho > 0.82$ (`\n\n`)
> * **Qwen2.5-7B-Instruct:** $\rho = 0.89$ (NLTK) and $\rho = 0.77$ (`\n\n`)
>
> We also added new heuristic baselines (selecting the response with the **Most Steps** or **Least Steps** based on NLTK splits) to verify that LALP is not simply acting as a proxy for verbosity.
>
> | Selection Method | 32B Avg Math | 7B Avg Math |
> | :--- | :--- | :--- |
> | **LALP (GLM-4.5-Air - Ours)** | **0.726** | **0.440** |
> | LALP (NLTK) | 0.719 | 0.431 |
> | LALP (`\n\n`) | 0.712 | 0.424 |
> | GALP (Baseline) | 0.632 | 0.412 |
> | Random (Baseline) | 0.651 | 0.407 |
> | Most Steps (Baseline) | 0.686 | 0.392 |
> | Least Steps (Baseline) | 0.642 | 0.408 |
>
> **Analysis:** The simple NLTK-based splitting performs remarkably close to the GLM-4.5-Air model and consistently outperforms both GALP and Random selection. The double newline (`\n\n`) method drops slightly in performance because different teacher models use structural spacing with varying frequencies, causing the split hierarchy to be unreliable across sources. Ultimately, these results prove that simpler approaches work highly effectively with LALP. The core mechanism, local context splitting, is robust and does not strictly depend on a specific LLM segmenter.
>
> ---
> ## **W2 & Q2: Context window generalization across domains**
>
> To confirm that our optimal context window size is not overfit to the mathematical domain, we extended our context window ablation to the coding domain. We generated responses for 5,000 prompts from the OpenCodeReasoning and LeetCode datasets, used them to fine-tune the Qwen models, and evaluated them on LiveCodeBench.
>
> | Context Window Size ($k$ steps) | 32B LCB Avg | 7B LCB Avg |
> | :--- | :--- | :--- |
> | LALP ($k=1$) | 0.561 | 0.334 |
> | LALP ($k=2$) | 0.579 | 0.342 |
> | LALP ($k=3$) | 0.578 | 0.345 |
> | LALP ($k=4$) | **0.589** | 0.350 |
> | LALP ($k=5$) | 0.576 | **0.352** |
> | GALP (Global) | 0.555 | 0.310 |
> | Random | 0.527 | 0.291 |
>
> **Analysis:** The optimal local window range ($k=4$ to $5$ steps) remains remarkably consistent with the mathematical domain. Expanding the window too far or shrinking it to just 1 step degrades performance back toward the GALP baseline. This demonstrates that the LALP context boundary can be a generalized property of step-compositional reasoning. We will add these cross-domain ablation curves to the paper.
>
> ---
> ## **W3 & Q3: Applicability during the RL stage**
>
> We note that the SFT + RL pipeline is the current vanguard for reasoning models. While our paper explicitly scopes itself to the SFT data curation bottleneck, LALP has direct and profound implications for the RL stage.
>
> Specifically, LALP acts as an implicit, zero-shot **Process Reward Model (PRM)**. In modern RL pipelines (e.g., GRPO/PPO), training an explicit PRM to provide step-level rewards is notoriously expensive and data-hungry. Because LALP evaluates the local learnability and standalone justifiability of individual reasoning steps, it can be used either as a zero-shot dense reward signal during RL, or as an automated filter to curate high-quality, step-level data for training explicit PRMs. We will add a dedicated paragraph to the discussion section to explicitly bridge this gap and highlight LALP's potential within the RL paradigm.
>
> ---
> ## **W4: Presentation, figures, and typos**
>
> We sincerely apologize for the formatting issues and the broken sentences. We have implemented the following presentation fixes for the revised manuscript:
> 1. We have replaced all bitmap figures with scalable vector graphics (.pdf) for clear plots.
> 2. We have increased the font sizes in Table 2, Table 3, and Figure 2 for better legibility.
> 3. We have corrected the incomplete sentence in Section 6 to read: "...aggregate at the step level. We formalize this as Local Average Log Probability."
> 4. We have fixed the LaTeX formatting for the LocalLP formulation and Equation 4, as well as the broken references.

---

> > ### Author Rebuttal · Reviewer_Z7Ai · 2026-04-04
> >
> > The authors addressed my core concerns well. The newly added ablation experiments on rule-based segmentation (NLTK) increased the credibility of the experiments and made the conclusions more reliable. Furthermore, the cross-domain ablation on LiveCodeBench demonstrated the generalizability of the optimal context window size. Regarding the applicability in the RL phase, the authors conceptually discussed its potential as a zero-shot PRM. Additionally, the authors have rectified all presentation and formatting issues. Overall, given the solid empirical evidence added for the SFT data curation pipeline and the practical utility of this approach, I am raising my score to Accept.

---

### Official Review · Reviewer_JE4e · 2026-03-16

**Soundness:** 3
**Presentation:** 2
**Significance:** 3
**Originality:** 3
**Overall Recommendation:** 4
**Confidence:** 3

**Summary:**

This paper introduces Local Average Log Probability (LALP) for response selection in the case of knowledge distillation from multiple stronger teacher models. They compare their method with Global Average Log Probability (GALP) and show that their approach performs better than GALP. They also analyze the mechanisms behind LALP's gains compared to GALP.

**Compliance With Llm Reviewing Policy:**

Affirmed.

**Ethical Review Concerns:**

No ethical concerns.

**Final Justification:**

I will keep my score.

**Key Questions For Authors:**

- How sensitive is LALP to the quality of step segmentation? Have you compared using simpler heuristics (e.g., splitting on "\n\n" or sentence boundaries) versus the LLM-based segmentation with GLM-4.5-Air? If simpler methods yield comparable results, this would strengthen the practical applicability of LALP; if not, it raises concerns about the external dependency.
- Have you tried all four student models and five teacher models listed in the Appendix? If so, what are the additional results. Does LALP still hold when testing on this broader set of student models and teacher models?
- The text states the self-conditioning gap is "near zero" at 0-10% position (Line 394), but the figure shows values over 0.7. Similarly, Lines 433-434 claim Qwen3 has the largest gap, but QwQ appears higher. Could you clarify these apparent discrepancies?
- Have you tried other simpler baselines such as selection based on response length or step count?
- Have you tried training on prompts other than those from the math domain? Would LALP selection generalizes to those domains as well?

**Limitations:**

Yes, the paper has added limitations in their Impact Statement.

**Strengths And Weaknesses:**

## Strengths
- They have motivated their approach well. They start from the failure cases from GALP and propose a new method to better capture step-compositional generalization.
- They have tested two scenarios, both very practical. One is selecting a teacher before fine-tuning, and the second is selecting a response for each prompt from a pool of responses generated by multiple teachers. They have demonstrated the effectiveness of their approach in both settings.
- They have added practical considerations given their approach requires more compute than the GALP method in line 259-269.
- They have also shown the cross-domain generalization by training on math prompts only but tested on other general science and coding benchmarks. The results show some performance gains from those benchmarks as well.
- They added plenty of analysis why GALP fails and what are the mechnisms that LALP fixes GALP's shortcomings.


## Weaknesses
- There are some inconsistencies in the paper. For example, in Appendix B.3, there are four student models listed and there are five teacher models listed. However, the primary results (Table 2 and Table 3) in this paper are only with two student models (Qwen2.5-7B-Instruct and Qwen2.5-32B-Instruct) and three teacher models ( Qwen3-32B, QwQ-32B, DeepSeek-R1). The supplementary results on Llama-3.1-8B-Instruct (Table 15) are helpful but relegated to the appendix and not sure which scenario it is tested on (coarse teacher selction or fine-grained response selection). This raises questions about whether the findings generalize across a broader range of model families and scales.
- The primary experiments use only LIMO prompts (817 examples) for training. Although the authors show transfer to GPQA-Diamond and LiveCodeBench, it would strengthen the paper to demonstrate LALP's effectiveness when training on more diverse reasoning datasets from the outset, rather than relying solely on cross-domain transfer experiments.
- LALP requires partitioning responses into reasoning steps using GLM-4.5-Air,  and this introduces an external dependency whose quality could vary across domains or teacher styles. The paper does not ablate the impact of segmentation quality on LALP's effectiveness, nor does it compare alternative segmentation methods (e.g., using "\n\n").
- Limited baselines. The paper primarily compares LALP to GALP and random selection. Other potential baselines worth considering include selection based on response length or step count.
- Presentations:
    - There are some typos:
        - Line 808: "Equation ??"
        - Line 880: "Equation ??"
        - Line 1079: "Section ??"
    - Line 394: "At the beginning of responses (0-10%), the gap is near zero" but I don't see this in the figure. Is this a typo? And in line 433-line 434: "Qwen3 shows the largest self-conditioning gap at all positions", but according to Figure 5, it seems that QwQ has the largest self-conditioning gaps. Please explain.

---

> ### Author Rebuttal · Authors · 2026-03-31
>
> Thank you for your highly constructive feedback. We have addressed your concerns regarding model inconsistencies, segmentation dependencies, baselines, and presentation discrepancies below.
>
> ---
> ## **W1 & Q2: Model inconsistencies and testing on a broader set of models**
>
> To clarify the model counts: the 3 reasoning teachers (DeepSeek-R1, QwQ-32B, Qwen3-32B) form our core mixed-teacher evaluation for long reasoning. The 2 non-reasoning teachers in the Appendix (Qwen2.5-72B, Gemma3-27B) generate much shorter, non-CoT responses and were used specifically for our sanity-check experiments. Llama-3.1-8B-Instruct was tested to demonstrate cross-architecture generalizability. We will clarify this setup and move the Llama-3.1 results to the main text.
>
> To strengthen our findings, we expanded the core experiment to combine **all 5 teacher models** into a single, heterogeneous response pool:
>
> | Selection Method | 32B Student (Avg Math) | 7B Student (Avg Math) |
> | :--- | :--- | :--- |
> | **LALP (Ours)** | **0.726** | **0.440** |
> | GALP  | 0.641 | 0.405 |
> | Random  | 0.594 | 0.389 |
>
> As shown, expanding the pool to 5 teachers actually causes the performance of both Random and GALP selection to drop further, as the candidate pool becomes more heterogeneous and noisy. In contrast, our method (LALP) successfully filters through this increased noise, maintaining strong and consistent performance. This confirms that LALP reliably selects high-quality reasoning data even as the candidate pool expands.
>
> ---
> ## **W2 & Q5: Training on non-math domains**
>
> We agree that evaluating on diverse datasets can strengthen the paper. We did, in fact, train on non-math domains for our coding evaluation. As detailed in Appendix C.10 (Table 14), we selected training data from a pool of 5,000 coding prompts (OpenCodeReasoning and LeetCode) and evaluated the resulting fine-tuned Qwen2.5-32B-Instruct model on LiveCodeBench v2. LALP successfully generalized to this domain, outperforming GALP (e.g., 0.261 vs. 0.232 on LiveCodeBench-Hard). We will refer a summary of these results to the main text to better highlight LALP's cross-domain applicability.
>
> ---
> ## **W3, W4, Q1 & Q4: Segmentation ablation and simple heuristic baselines**
>
> We conducted ablations using two simple, rule-based splitting heuristics: Python's **NLTK sentence tokenizer** (with LaTeX post-processing) and double newline (`\n\n`) splitting.
>
> Additionally, we added two new verbosity-based baselines: selecting the response with the **Most Steps** and the **Least Steps** (calculated via NLTK splits). Other recent likelihood-based selection methods (e.g., Jung et al. 2025; Panigrahi et al. 2025) use gradient information to select teacher models rather than individual responses, making them computationally intractable for 10K+ token reasoning traces and not suitable for our setting.
>
> | Selection Method | 32B Avg Math | 7B Avg Math |
> | :--- | :--- | :--- |
> | **LALP (GLM-4.5-Air - Ours)** | **0.726** | **0.440** |
> | LALP (NLTK) | 0.719 | 0.431 |
> | LALP (`\n\n`) | 0.712 | 0.424 |
> | Most Steps (Baseline) | 0.686 | 0.392 |
> | Random (Baseline) | 0.651 | 0.407 |
> | Least Steps (Baseline) | 0.642 | 0.408 |
> | GALP (Baseline) | 0.632 | 0.412 |
>
> 1. **Segmentation:** LALP rankings using NLTK and `\n\n` showed strong Spearman correlations (>0.82 to >0.92) with the GLM-4.5-Air splits. While the LLM provides the cleanest semantic boundaries, NLTK-based splitting performs remarkably well. The `\n\n` method is slightly less effective because different teacher models use structural spacing with varying frequencies, causing the split hierarchy to be unreliable across teachers. However, simpler approaches still yield a highly meaningful selection signal, proving LALP does not strictly depend on an external LLM.
> 2. **Baselines:** LALP consistently outperforms simple verbosity heuristics (Most Steps and Least Steps) across both student models. Selecting simply for "Most Steps" falls into the same fluency trap as GALP, rewarding verbose scaffolding over concise, accurate logic.
>
> ---
> ## **W5 & Q3: Presentation issues and Figure 5 discrepancies**
>
> We apologize for the broken LaTeX references (`??`), which have been corrected.
>
> * **Figure 5:** The plot legend colors were inadvertently swapped. Qwen3-32B should be Blue, and QwQ-32B should be Red. Fixing this typo aligns the plot perfectly with the text stating that Qwen3 exhibits the largest self-conditioning gap.
> * **Line 394:** When we stated the gap is "near zero" at the 0-10% position, we were referring to the variance between the different teacher models, not the absolute self-conditioning gap itself. We realize this phrasing was highly ambiguous and will rewrite this section clearly to distinguish between the inter-teacher gap and the self-conditioning gap.

---

> > ### Author Rebuttal · Reviewer_JE4e · 2026-04-04
> >
> > Thanks for the author response, I will keep my score.

---

### Official Review · Reviewer_yRVM · 2026-03-19

**Soundness:** 3
**Presentation:** 2
**Significance:** 3
**Originality:** 3
**Overall Recommendation:** 4
**Confidence:** 4

**Summary:**

This paper proposed a new data selection method for SFT on reasoning trace data, trying to answer the question: Which reasoning trace should be selected as training data for the student model?

Old methods, like GRAPE, choose the response that the student model assigns the highest probability and select the response that best fits the student’s pretrained distribution (the most natural w.r.t. the student model), and this paper reveals limitation of this method (we call it "old method" from now on) and proposes its improved method (LALP).

The key insight is that reasoning generalizes at the step level rather than the full trajectory level, so selection should evaluate step-level learnability instead of global fluency.

Instead of evaluating the likelihood of the entire response, LALP:

Splits the reasoning trace into steps, scores each step using the student model (in terms of likelihood). There are 2 main differences:

1. Each step is conditioned only on a small local context window: Let k be the window size, the log-likelihood score of each token is only calculated conditioned on the previous k steps and the current step, instead of the entire previous tokens in the old method

2. Averages over the step scores instead of token scores: A "step score" is the average of the token log-likelihood score within that step. The paper choose to average over step scores, essentially treats each step as equal, instead of the old method, which average over token score, which implicitly assigns more weights to longer steps.

In the evaluation part of the paper, the authors show that this method outperforms the old method in both coarse-grained teacher model selection and fine-grained per-prompt response selection from a mixed-teacher pool.

The paper also explained the insight on why the old method had such limitation and why LALP fixed it. It gives 2 main arguments:

1. Information-rich reasoning-critical token are often locally low-probability, while filler words are highly predictable. For long reasoning, fluent but non-informative sentence may receive higher score than high reasoning quality traces due to the fact that old method average over all tokens' scores.

2. Later reasoning tokens are very easy to predict if conditioned on the entire reasoning trace, not necessary because the reasoning itself is good, but that they are consistent with earlier text.

**Compliance With Llm Reviewing Policy:**

Affirmed.

**Final Justification:**

The rebuttal addressed my main concerns. Overall I think this paper's contribution overweights the weakness, as long as the formats and figures are fixed in the future version of the paper. I therefore retain my positive score with more confidence.

**Key Questions For Authors:**

1. Regarding to the SFT data selection method, is there any other method, other than the GRAPE (GALP) discussed in the paper? In this case, do you think it's necessary to also include this in your experiment for comparison?

**Limitations:**

Mostly covered by the paper's limitation section. An additional concern is raised as question 1 in the above section

**Strengths And Weaknesses:**

## Strength

The main strength of the paper lies in the evaluation (mainly 7.2) as well as the mechanics explanation. The details are already covered in he paper summary mentioned above.


## Weakness

### Experiment design
In 7.1 (teacher model selection), as well as figure 1, the experiment only takes 3 teacher models and check if the overall GALP/LALP rankings align with the aveage performance. Essentially, if the selection criteria is random, there is a 1/6 chance that the rankings still align, so it's not convincing. It would be sounder if more teacher models (a total of 5) are used in the experiment.

### Paper Presentation
There are major problems in paper presentation.

Figure 1, 2, 3 and 6 are bitmaps, which should not be used in plotting curves in a research paper in most cases. In this specific case, these figures are regular line plot, so please re-plot these figures in the same format as other figures, and make sure to export these figures as vector graphics (like pdf)

In the reference part, some arxiv papers have a url of https://www.semanticscholar.org/ , some under https://arxiv.org/ and others' url are missing. These reference should be fixed to make it consistent:

1. If the paper is accepted in any conference/journal, please cite the conference/journal instead of arxiv, unless there are significant changes between these versions

2. Otherwise, please include the url of https://arxiv.org/ . There is no point using a third-party api like semantic scholar.

Overall, the contribution overweights the weakness of the paper and I'm leaning to accept this paper. However, the issue mentioned in the "Paper presentation" must be fixed at the end of the discussion period.

---

> ### Author Rebuttal · Authors · 2026-03-31
>
> Thank you for your supportive assessment of our work and for the highly constructive feedback. We have addressed all of your concerns below and in the revised manuscript.
>
> ---
> ## **W1: Experiment design and the number of teacher models**
>
>
> In our original submission, we focused on 3 specific teachers due to their ability to output long reasoning in mathematical tasks. To solidify our claims, we have now expanded the mixed-teacher response selection pool to include a total of 5 diverse teacher models (adding Qwen2.5-72B-Instruct and Gemma3-27B-IT). The updated downstream performance for the student models is shown below:
>
> | Selection Method | Qwen2.5-32B-Instruct (Avg Math Acc) | Qwen2.5-7B-Instruct (Avg Math Acc) |
> | :--- | :--- | :--- |
> | **LALP (Ours)** | **0.726** | **0.440** |
> | GALP  | 0.641 | 0.405 |
> | Random  | 0.594 | 0.389 |
>
> As shown, expanding the pool to 5 teachers actually causes the performance of both Random and GALP selection to drop further, as the candidate pool becomes more heterogeneous and noisy. In contrast, our method (LALP) successfully filters through this increased noise, maintaining strong and consistent performance. This confirms that LALP reliably selects high-quality reasoning data even as the candidate pool expands.
>
> ---
> ## **W2: Paper Presentation (Figures and References)**
>
> We apologize for the formatting issues and have implemented all of your suggested presentation fixes for the revision:
> 1. **Figures:** We have re-plotted Figures 1, 2, 3, and 6 and exported them strictly as scalable vector graphics (.pdf) to ensure they are clear at any level.
> 2. **References:** We have conducted a comprehensive pass of the bibliography. We removed all third-party API links (e.g., Semantic Scholar), standardized all preprint URLs to official arXiv links, and updated all accepted papers to cite their official conference or journal proceedings rather than their preprint versions.
>
> ---
> ## **Q1: Comparison with other SFT data selection methods**
>
> Our core focus is student-aware, response-level selection. In this specific paradigm, GALP (via GRAPE, NeurIPS 2025) is the state-of-the-art baseline for selecting responses from a multi-teacher pool using likelihood.
>
> Other recent data selection methods, such as those by Jung et al. (2025) or Panigrahi et al. (2025), utilize gradient information and focus on selecting teacher models rather than scoring individual responses. This makes them unsuitable for our setting and expensive for 10K+ token reasoning traces. Our primary goal is to demonstrate that relying on a single global metric (GALP) can be actively misleading for long reasoning, and that shifting to local evaluation yields a stronger, more accurate signal for the student.
>
> However, to further strengthen our evaluation and provide additional baselines beyond GALP and Random, we have introduced additional heuristic-based selection: selecting the response with the **Most Steps** and the **Least Steps** (calculated using NLTK tokenizer splits).
>
> | Model / Selection Method | Downstream Average Math Performance |
> | :--- | :--- |
> | **Qwen2.5-32B-Instruct** | |
> | LALP (GLM-4.5-Air) | **0.726** |
> | Most Steps | 0.686 |
> | Least Steps | 0.642 |
> | **Qwen2.5-7B-Instruct** | |
> | LALP (GLM-4.5-Air) | **0.440** |
> | Least Steps | 0.408 |
> | Most Steps | 0.392 |
>
> As shown above, LALP consistently outperforms these verbosity-based heuristics. Selecting simply for the "Most Steps" often falls into the same fluency trap as GALP, rewarding verbose scaffolding over concise, accurate logic. This confirms that LALP successfully optimizes for step-level reasoning quality rather than mere volume.

---

> > ### Author Rebuttal · Reviewer_yRVM · 2026-04-03
> >
> > I thank the author for the reply. My main concerns are resolved.

---

### Official Review · Reviewer_YnHD · 2026-03-20

**Soundness:** 2
**Presentation:** 3
**Significance:** 3
**Originality:** 3
**Overall Recommendation:** 4
**Confidence:** 3

**Summary:**

The authors claim that global response likelihood is a bad criterion for selecting reasoning data from multiple teachers. Instead, they propose LALP which scores each reasoning step via a small window of prior context. Empirically, LALP outperforms prior approaches in teacher ranking and prompt selection, and leads to stronger student models across different parameter scales.

**Compliance With Llm Reviewing Policy:**

Affirmed.

**Final Justification:**

The authors’ rebuttal has addressed most of my concerns to some extent.

**Key Questions For Authors:**

- Based on the experimental setup the method picks only among correct candidates. How will it perform when incorrect responses are also included?
- Any reason you don't compare with any other stronger baselines compared to random and GALP?

**Limitations:**

Yes

**Strengths And Weaknesses:**

**Strengths**
- The proposed method is straightforward, ie step-level local scoring seems like a sensible modification of GALP.
- Empirically LALP seems to succeed where approaches like GALP fail.
- Informative analyses such as context window-size ablation, token-attribution analysis, and self-conditioning analysis.

**Weaknessess**
- The method depends arbitrarily on an auxiliary model for segmentation without an ablation showing how more simpler approaches can perform.
- Experimental setup seems quite narrow, eg selection is only correct candidates from a limited pool.
- Gains are uneven across students. For the 7B student LALP outperforms GALP but with very small improvement over random  compared to 32b student.

---

> ### Author Rebuttal · Authors · 2026-03-31
>
> Thank you for your constructive feedback. We have addressed your concerns below and updated our empirical results accordingly.
>
> ---
> ### **W1: Dependence on an auxiliary model for segmentation**
>
> We agree that minimizing external dependencies is important. We tested simpler, rule-based segmentation strategies: Python's NLTK tokenizer (with LaTeX post-processing) and double newline (`\n\n`) splitting. Both showed strong Spearman correlations with our GLM-4.5-Air rankings:
> * **Qwen2.5-32B:** $\rho = 0.92$ (NLTK), $\rho = 0.82$ (`\n\n`)
> * **Qwen2.5-7B:** $\rho = 0.89$ (NLTK), $\rho = 0.77$ (`\n\n`)
>
> As shown in the table below, while GLM-4.5-Air provides the most meaningful semantic boundaries and the highest performance, the simpler NLTK-based splitting performs remarkably well and still cleanly outperforms the GALP and Random baselines. The double newline (`\n\n`) method is slightly less reliable because different teacher models use spacing with varying frequencies, but it still yields a meaningful selection signal. We will add these ablation results to the manuscript to demonstrate that while a capable LLM boosts performance, LALP's core mechanism, local context truncation, works effectively even with simpler heuristic splitting.
>
>
> | Selection Method | 32B Avg Math | 7B Avg Math |
> | :--- | :--- | :--- |
> | **LALP (GLM-4.5-Air)** | **0.726** | **0.440** |
> | LALP (NLTK) | 0.719 | 0.431 |
> | LALP (`\n\n`) | 0.712 | 0.424 |
> | GALP (Baseline) | 0.632 | 0.412 |
> | Random (Baseline) | 0.651 | 0.407 |
>
> ---
> ### **W2+Q1: Performance when incorrect candidates are included**
>
> In reasoning tasks (math, code), verifying the final answer via rule-based execution (e.g., SymPy, unit tests) is standard, cheap, and highly scalable. Consequently, SFT data selection is typically bottlenecked by supervision quality rather than correctness verification, which is why methods like DeepSeekMath, OpenThoughts, and our dataset all filter for correctness first.
>
> However, to address how LALP performs when incorrect responses are included, we evaluated it directly on an unfiltered code dataset. We selected data from a pool of 5,000 prompts (OpenCodeReasoning and LeetCode) and evaluated it on LiveCodeBench (detailed in Appendix C.10, Table 14).
>
> | Selection Method | LCB-Easy | LCB-Medium | LCB-Hard |
> | :--- | :--- | :--- | :--- |
> | Original Qwen2.5-32B | 0.890 | 0.471 | 0.114 |
> | GALP / GRAPE | 0.845 | 0.588 | 0.232 |
> | **LALP (Ours)** | **0.874** | **0.633** | **0.261** |
>
> Theoretically, if forced to score incorrect traces, LALP is actually more robust than GALP. As formalized in our "fluency trap" theorem (Theorem A.2), incorrect responses often mask logical errors behind highly predictable, fluent discourse tokens. GALP averages over the whole document, allowing high-probability "fluff" to outweigh low-probability math errors. LALP isolates the local step, making it much harder for a hallucinated leap to hide behind global stylistic coherence.
>
> ---
> ### **W3: Uneven gains across 7B and 32B students**
>
> The performance difference reflects the capacity limits of 7B-class models, rather than a flaw in the selection metric. Achieving meaningful reasoning improvements in 7B models typically requires massive datasets (e.g., AceReason-Nemotron-1.1-7B uses 2.2M samples [1]; OpenThinker3-7B uses 1.2M [2]).
>
> Crucially, even with this severe capacity and data bottleneck, LALP still consistently outperforms both the Random and GALP baselines on the Qwen2.5-7B-Instruct model (as well as the Llama-3.1-8B-Instruct model, shown in Appendix C.11). The fact that LALP extracts a reliable, accurate selection signal from only smaller examples shows the effectiveness of data-efficiency of our method.
>
> > [1] Liu, Zihan, et al. "Acereason-nemotron 1.1... 2025."
> > [2] Guha, Etash, et al. "Openthoughts: Data recipes... 2025."
>
> ---
> ### **Q2: Comparison with stronger baselines**
>
> We focus on student-aware, zero-shot, response-level selection, where GALP is the state-of-the-art baseline. Recent gradient-based methods (Jung et al., 2025; Panigrahi et al., 2025) select *teacher models* rather than individual responses, making them not appropriate for your setting for response selection and prohibitive for computation on long reasoning data.
>
> However, to further strengthen our evaluation, we have introduced simple heuristic baselines: selecting the response with the **Most Steps** and the **Least Steps** (calculated using NLTK splits).
>
> | Selection Method | 32B Avg Math | 7B Avg Math |
> | :--- | :--- | :--- |
> | **LALP (GLM-4.5-Air)** | **0.726** | **0.440** |
> | Most Steps | 0.686 | 0.392 |
> | Least Steps | 0.642 | 0.408 |
>
> As shown above, LALP consistently outperforms these verbosity-based heuristics. Selecting simply for the "Most Steps" often falls into the same fluency trap as GALP, rewarding verbose scaffolding over concise, accurate logic. This confirms that LALP successfully optimizes for step-level reasoning quality rather than mere volume.

---

> > ### Author Rebuttal · Reviewer_YnHD · 2026-04-04
> >
> > Thank you for your response. I will increase my score.

---

### Decision · Program_Chairs · 2026-04-30

**Decision:**

Accept (spotlight)

**Comment:**

This paper presents an important and timely method to select reasoning data for language model development.
I agree with all reviewers that this is a valuable contribution and strongly recommend this paper be accepted.

Action item for authors: please revise the manuscript as promised to reviewers, including reorganizing the text and formatting of figures.